# Modulation of telomere protection by the PI3K/AKT pathway

Marinela Méndez-Pertuz[1], Paula Martínez[1], Carmen Blanco-Aparicio[2], Elena Gómez-Casero[2], Ana Belen García[2], Jorge Martínez-Torrecuadrada[3], Marta Palafox[4], Javier Cortés[4], Violeta Serra[4], Joaquin Pastor[2] & Maria A. Blasco[1]

Telomeres and the insulin/PI3K pathway are considered hallmarks of aging and cancer. Here, we describe a role for PI3K/AKT in the regulation of TRF1, an essential component of the shelterin complex. PI3K and AKT chemical inhibitors reduce TRF1 telomeric foci and lead to increased telomeric DNA damage and fragility. We identify the PI3Kα isoform as responsible for this TRF1 inhibition. TRF1 is phosphorylated at different residues by AKT and these modifications regulate TRF1 protein stability and TRF1 binding to telomeric DNA in vitro and are important for in vivo TRF1 telomere location and cell viability. Patient-derived breast cancer PDX mouse models that effectively respond to a PI3Kα specific inhibitor, BYL719, show decreased TRF1 levels and increased DNA damage. These findings functionally connect two of the major pathways for cancer and aging, telomeres and the PI3K pathway, and pinpoint PI3K and AKT as novel targets for chemical modulation of telomere protection.

[1] Telomeres and Telomerase Group, Molecular Oncology Program, Spanish National Cancer Centre (CNIO), Melchor Fernández Almagro 3, Madrid E-28029, Spain. [2] Experimental Therapeutics Program, Spanish National Cancer Centre (CNIO), Melchor Fernández Almagro 3, Madrid E-28029, Spain. [3] Biotechnology Program, Spanish National Cancer Centre (CNIO), Melchor Fernández Almagro 3, Madrid E-28029, Spain. [4] Experimental Therapeutics Group, Vall d´Hebron Institute of Oncology (VHIO), Natzaret 115-117, Barcelona E-08035, Spain. Marinela Méndez-Pertuz and Paula Martínez contributed equally to this work. Correspondence and requests for materials should be addressed to M.A.B. (email: mblasco@cnio.es)

Telomeres are nucleoprotein structures located at the ends of eukaryotic chromosomes that protect them from degradation and DNA repair activities, ensuring chromosomal stability[1]. One of the hallmarks of cancer is the ability of cancer cells to maintain functional telomeres, mainly through the activation of telomerase, the enzyme that elongates telomeres, thus allowing for telomere maintenance and indefinite cancer cell division[2, 3]. Targeting telomeres in cancer has been a goal for the past decades. However, telomerase inhibition in cancer cells only showed effectiveness in some myeloid tumors and has largely failed in other tumor types[4–6]. This is likely due to the fact that telomerase is not essential for cell division until telomeres reach a critically short length, together with the fact that tumors contain heterogeneous cell populations also with regard to telomere length[7–10]. In addition, alternative mechanisms to elongate telomeres known as ALT can be activated in telomerase-negative tumors[4–6].

In vertebrates, telomeres are composed of tandem repeats of the TTAGGG sequence bound by a six protein complex known as shelterin, which encompasses TRF1, TRF2, TIN2, POT1, RAP1, and TPP1[11, 12]. The telomere repeat binding factor 1 (TRF1) was the first shelterin to be discovered[13] and it is one of the best characterized. TRF1 binds to telomeric DNA as a homodimer, thus protecting telomeres[14, 15]. A number of reports have shown that binding of human TRF1 to telomeres can be regulated post-translationally. Among these modifications, polyADP ribosylation of TRF1 is the best understood. ADP-ribosylation of human TRF1 by tankyrase 1 releases TRF1 from telomeres promoting its ubiquitination and proteasome-mediated degradation[16–20]. Other modifications include phosphorylation by polo-like kinase 1 (PLK1), cyclin B-dependent kinase 1 (CDK1), casein kinase 2 (CK2), a member of the never in mitosis gene A (NIMA) kinase family Nek7, and by AKT[20–25]. However, functional relevance of these modifications is poorly understood. TRF1 is also regulated at the transcriptional level by the pluripotency factor Oct4, which binds to TRF1 promoter and it is sufficient to highly upregulate TRF1 messenger RNA (mRNA) and protein levels during in vitro generation of induced pluripotent stem cells (iPS cells)[26]. Indeed, TRF1 has an essential role both in the induction and maintenance of the pluripotency state[26]. In agreement with this, genetic depletion of TRF1 leads to mouse lethality at the blastocyst stage[27].

In the context of differentiated tissues, genetic deletion of TRF1 in adult mice results in loss of telomere protection, increased telomere fragility, and activation of a persistent DNA damage response (DDR) at chromosome ends, which leads to cellular senescence and/or apoptosis and to severe impairment of tissue regeneration and organismal viability[26, 28–31]. All these effects occur in the absence of telomere shortening, indicating that TRF1 inhibition is sufficient to induce severe telomere dysfunction in the absence of telomere shortening. Thus, shelterins could be potential therapeutic targets to rapidly disrupt telomeres in cancer cells independently of telomere length, in this manner overcoming the limitations of telomerase inhibition. Indeed, similar to telomerase, some shelterin components have been recently found mutated both in familial and sporadic human tumors, such as mutations in the TRF1-interacting protein POT1[32–38]. Furthermore, we recently demonstrated that genetic deletion of TRF1 in a mouse lung cancer model impairs the growth of p53-null K-RasG12V-induced lung carcinomas and increases mouse survival independently of telomere length[39]. In particular, TRF1 genetic deletion impaired cancer growth concomitant with induction of telomeric DNA damage, apoptosis, decreased proliferation, and G2-arrest[39]. These phenotypes associated to TRF1 genetic deletion were also achieved by using small molecule inhibitors of TRF1 telomeric foci[39]. These TRF1 inhibitors were identified in a phenotypic screening to disrupt TRF1 foci using a curated library (ETP-640) representing the chemical space of our in-house 50,000 compound ETP-CNIO collection[39].

The phosphatidylinositol-3-kinase (PI3K) pathway regulates a wide range of target proteins to control cell proliferation, survival, and cell growth[40]. There are three major classes of PI3K enzymes, being class IA widely associated to cancer. Class IA PI3K are heterodimeric lipid kinases composed of a catalytic subunit (p110α, p110β, or p110δ; encoded by PIK3CA, PIK3CB, and PIK3CD genes, respectively) and a regulatory subunit (p85)[41]. Following extracellular mitogenic stimulation, activated p110 subunit catalyzes the production of phosphatidylinositol-3,4,5-triphosphate, a lipid second messenger, that in turn activates the serine/threonine protein kinase AKT/protein kinase B (PKB) and other effectors[41]. A number of mutations in the PI3K/ATK pathway have been found in human cancers[42, 43], thus targeting the PI3K pathway is a therapeutic strategy in cancer[44]. Although many PI3K downstream targets are proposed to mediate the role of PI3K in cancer[45, 46], none of them have been linked to maintenance of chromosomal stability or to telomere function, which are hallmarks of cancer[47].

In analogy to telomeres, PI3K also has a fundamental role in aging and longevity[48, 49]. In particular, PI3K is downstream of the glucose-dependent and of IGF-1 signaling pathways and its downregulation, as well as downregulation of its downstream kinases mTOR and S6K is shown to extend life-span in several organism from yeast to mice[48, 50–54].

As mentioned above for TRF1[26], the PI3K pathway also plays crucial role in maintenance of pluripotency and in the survival of iPS cells (IPSCs)[55, 56]. Indeed, reprogramming efficiency is enhanced by the combined chemical inhibition of a PI3K downstream target, namely GSK3-β and a RAS downstream target MEK/ERK[57]. In particular, the glycogen synthase kinase (GSK3) is downstream of PI3K/AKT and is repressed upon PI3K-AKT signaling[58]. A connection between TRF1 and the PI3K pathway in the maintenance of the pluripotency state is unknown to date.

In this report, we make the discovery that the TRF1 telomeric protein is regulated by the PI3K signaling pathway. In particular, inhibition of PI3K and of its downstream target AKT by small molecules, as well as genetic depletion of the PIK3CA gene encoding the p110α catalytic subunit of PI3Kα resulted in decreased TRF1 protein levels and decreased TRF1 telomeric foci. This in turn lead to increased telomere fragility and increased telomere aberrations (i.e., multitelomeric signals). We further find that AKT phosphorylates purified TRF1 in vitro, suggesting a direct involvement of PI3K/AKT axis in the regulation of TRF1. In this regard, we identify three AKT phosphorylation sites in TRF1 at residues T248, T330, and S344. Overexpression of eGFP-tagged Trf1 mutants Trf1^T330A and Trf1^S344A rendered significantly lower TRF1 telomeric foci compared to cells expressing wild-type Trf1 both in lung tumor cells and in MEFs. In addition, while wild-type eGFP-tagged Trf1 was able to rescue proliferation of TRF1-deficient MEFs, this was not the case for eGFP-tagged Trf1 mutants Trf1^T330A and Trf1^S344A, thus demonstrating the relevance of these AKT-dependent TRF1 modifications for cell viability and TRF1 telomeric foci formation in vivo.

Finally, we extended these findings to the clinical setting by using patient-derived xenograft (PDX) models. We show that patient-derived breast cancer PDX models that responded to the treatment with a specific PI3Kα inhibitor also show significantly decreased TRF1 levels and increased telomeric DNA damage. Together, these findings demonstrate a central role of the PI3K/AKT pathway in regulation of telomere protection, thus highlighting components of this pathway as novel targets for telomere-based therapies in cancer and age-related diseases.

## Results

**The PI3K-AKT pathway regulates TRF1 telomeric foci.** By performing a high-throughput phenotypic screening designed to identify compounds that disrupt the location of eGFP-tagged TRF1 to telomeres, we previously identified the hit compounds ETP-47037 and ETP-47228 (proprietary CNIO compounds) as inhibitors of eGFP-TRF1 telomeric foci fluorescence (Fig. 1a, top)[39].

Here, we set to deconvolute the molecular pathways responsible for inhibition of TRF1 foci by these small molecules. We noted that both compounds belonged to a chemical series previously identified by us as PI3K inhibitors[59, 60], thus we set to address whether the PI3K pathway was able to regulate TRF1 foci formation. First, we confirmed that both ETP-47037 and ETP-47228 had a potent inhibitory activity against the PI3Kα isoform, with $IC_{50}$ values of 0.99 and 2.6 nM, respectively (Fig. 1b, "Methods" section). In addition, ETP-47037 also inhibited the PI3Kβ, PI3Kδ, and PI3Kγ isoforms, with $IC_{50}$ values of 49.2, 7.13, and 49.1 nM, respectively (Fig. 1b). Of interest, ETP-47037 did not inhibit significantly the PI3K downstream kinase mTOR kinase, ($IC_{50}$ of 1.7 μM), while ETP-47228 showed a moderate mTOR kinase inhibition ($IC_{50}$ of 443 nM) (Fig. 1b).

Next, to address whether PI3K activity was responsible for the inhibition of TRF1 foci, we designed two close analogs of the active compounds ETP-47037 and ETP-47228, ETP-51259 and ETP-50952, respectively, with a negligible activity against PI3K (Fig. 1a, b). These new chemical probes were prepared by replacement of a single key oxygen atom from the morpholine moiety by a methylene group, generating the piperidine derivatives ETP-51259 and ETP-50952 (Fig. 1a). To assess their PI3K cellular activity, we measured phosphorylation of residue

Ser473 in AKT (p-AKT S473), a PI3K downstream target[61–63], by western blot analysis. To this end, we used the CHA-9.3 mouse lung cancer cell line previously generated by us[39]. As expected, p-AKT S473 was strongly inhibited by our active compounds ETP-47037 and ETP-47228, but largely unaffected by their inactive versions (ETP-51259 and ETP-50952) (Fig. 1c). Of note, the inhibitory activity of ETP-47037 was more prolonged than that of ETP-47228. Thus, ETP-47037 mediated p-AKT S473 inhibition was maintained during the 24-hour treatment while the ETP-47228 mediated inhibition decreased 35% after 24 h as compared to 1-h treatment but still was able to inhibit the pathway above 50% (Fig. 1c).

We also determined the effect of ETP-47037 and ETP-47228 in the inhibition of other PI3K downstream targets, PRAS40 (T246), and pS6 (S235/236)[64, 65] (Supplementary Fig. 1). The results clearly show that both ETP compounds constitute bona fide inhibitors of the PI3K/AKT axis.

Next, we addressed whether the inactive PI3K/AKT compounds also failed to inhibit TRF1 foci when used in in vivo cellular assays. To this end, we treated CHA-9.3 cells with a 10 μM concentration of both the active and inactive chemical probes for 24 h and quantified TRF1 spot intensity by immunofluorescence (i.e., TRF1 shows a spotted pattern in interphase nuclei corresponding to telomeres)[29] ("Methods" section). Interestingly, cells treated with the inactive ETP-51259 and ETP-50952 compounds showed negligible inhibition of TRF1 foci, in contrast to efficient inhibition of TRF1 foci by the active compounds ETP-47037 and ETP-47228 (Fig. 1d), confirming that inhibition of PI3K/AKT is necessary for the activity of these compounds inhibiting TRF1 foci.

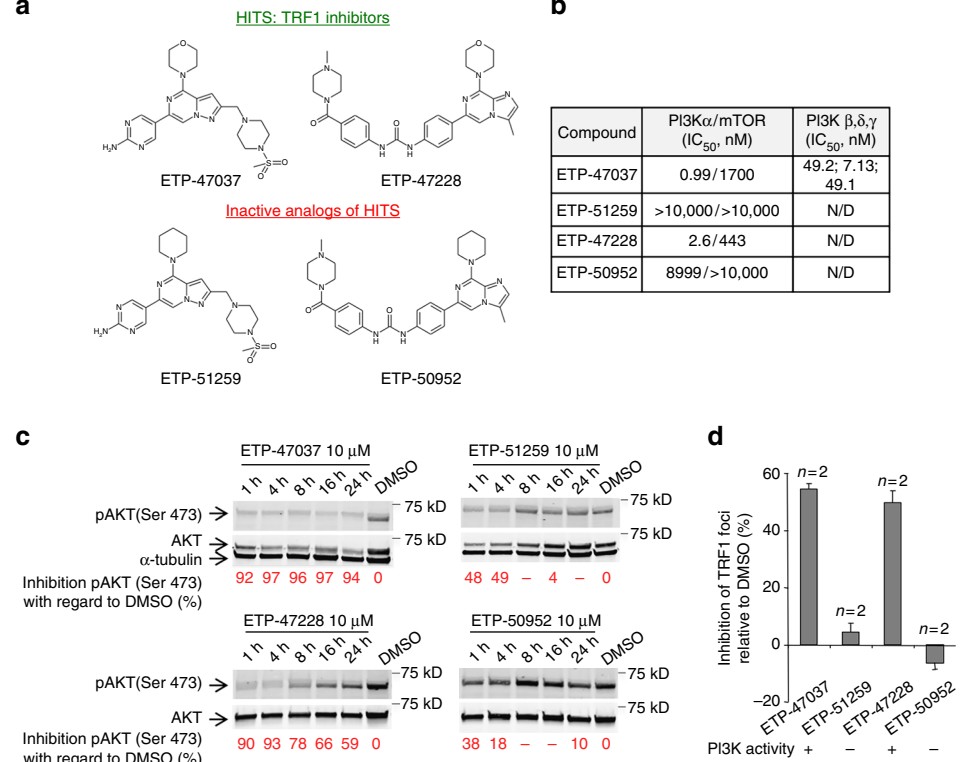

**Fig. 1** Chemical inhibition of TRF1 binding to telomere by PI3K inhibitors. **a** Structures of ETP-47037, ETP-47228, and their corresponding "inactive analogs" ETP-51259 and ETP-50952. **b** PI3K/mTOR $IC_{50}$ data generated internally and reported for the inhibitors used in the study. **c** Time course Inhibition of AKT phosphorylation at Ser473 by ETP-47037, ETP-51259, ETP-47228, and ETP-50952 at 10 μM in CHA-9.3 cell line. **d** Percent inhibition of TRF1 foci by immunofluorescence in CHA-9.3 mouse lung tumor cell line treated with 10 μM of either ETP-47037, ETP-47228, or their corresponding inactive analogs (ETP-51259 and ETP-50952, respectively) relative to TRF1 levels with DMSO treatment. The ETP compound inhibitory activity on PI3K pathway is stated at the bottom of the graph. Error bars represent standard deviation. N/D not determined, *n* number of independent experiments

As we observed that ETP-47228 also showed some inhibition of the PI3K/AKT downstream kinase mTOR ($IC_{50}$ of 443 nM; Fig. 1b), we set to address whether inhibition of TRF1 foci by these compounds was mediated by mTOR. To this end, we treated cells with a wide panel of available PI3K inhibitors, with or without activity against mTOR (Fig. 2a–d). We selected compounds sharing similar structural features to our hits, such as ETP-46992, another CNIO proprietary inhibitor[66] and GDC-0941[67], or with quite dissimilar structures such as BKM-120[68], BYL-719[69], TGX-221[70], BEZ-235[71], MK-2206[72], and GSK-2126458[73] (Fig. 2a). We determined the activity of the different compounds against the PI3Kα isoform and against mTOR by using biochemical assays for each enzyme, and obtained $IC_{50}$ values similar to those previously reported for these compounds (Fig. 2b) ("Methods" section). In particular, all the compounds inhibited PI3Kα at nanomolar level, except TGX-221, which showed negligible PI3Kα inhibition (Fig. 2c, d). Regarding mTOR inhibition, BEZ-235 and GSK-2126458 presented potent nanomolar mTOR inhibition, whereas GDC-0941, BKM-120 and ETP-46992, showed a micromolar activity against mTOR (Fig. 2b). Finally, the PI3Kα selective inhibitor BYL-719 did not inhibit mTOR ($IC_{50} > 10 \mu M$), and thus was used as a good chemical probe to test the influence of mTOR on TRF1 foci formation.

The kinase selectivity of the inhibitors used here was ensured in order to discard potential off-targets effects contributing to the observed TRF1 modulation. Thus, ETP-47037 and ETP-47228 were profiled against a panel of 24 kinases representing the human kinome, producing negligible inhibition of any kinase tested (Supplementary Table 1). Moreover, the rest of inhibitors used can be categorized as highly selective inhibitors of their PI3K/AKT pathway main targets (Fig. 2b) attending to published reports. The detailed selectivity information can be accessed in the references cited above and additional complementary studies: ETP-46992[66]; GDC-0941[67], and (http://lincs.hms.harvard.edu/db/datasets/20060; http://lincs.hms.harvard.edu/db/datasets/20261); BKM-120[68] and (http://lincs.hms.harvard.edu/db/datasets/20262); BYL-719[69, 74] and (http://lincs.hms.harvard.edu/db/datasets/20263); TGX-221[70] and (http://lincs.hms.harvard.edu/db/datasets/20223); BEZ-235[71] and (http://lincs.hms.harvard.edu/db/datasets/20096); and GSK-2126458[73] and (http://lincs.hms.harvard.edu/db/datasets/20084).

Next, we measured the inhibition of PI3Kα activity by these compounds in the CHA-9.3 cell line by determining the phosphorylation of AKT (p-AKT S473) at 24 h after treatment with 10 μM concentration of each compound, except for GSK-2126458, which was dosed at 1.0 μM due to its high cytotoxicity at 10 μM. As shown in Fig. 2c, phosphorylation of AKT in the CHA-9.3 cell line was strongly reduced by all these molecules with the exception of TGX-221, which only showed a moderate inhibition. In these conditions, all these PI3K inhibitors, independently of their structure and their mTOR activity, were able to inhibit TRF1 foci (Fig. 2d), confirming a role of the PI3K signaling pathway in TRF1 regulation, which is independent of mTOR inhibition. As a general trend, the strongest inhibition of p-AKT (S473) correlated with the highest downregulation of TRF1 foci, being the TGX-221 compound the weakest TRF1 inhibitor and also the one exerting the weakest p-AKT inhibition (Fig. 2d). Importantly, the specific inhibitor for the p110α isoform, BYL719, efficiently inhibited TRF1 foci (30%), whereas the specific inhibitor for the p110β isoform TGX221 only decreased by 10% TRF1 foci fluorescence (Fig. 2d). These results indicate that the PI3K isoform p110α is the main activity involved in TRF1 regulation at telomeres and that mTOR inhibition is not necessary.

We next studied the direct effect of AKT inhibition on TRF1 foci formation. To this end, we treated CHA-9.3 cells with the previously described allosteric AKT inhibitor MK-2206[72] at 10 μM concentration (Fig. 2a–d). Similarly to PI3Kα inhibitors, MK-2206 effectively inhibited TRF1 foci, which correlated with strong inhibition of p-AKT at Ser473 (Fig. 2c, d).

We then checked whether PI3Kα/AKT inhibition also leads to decreased TRF1 foci in an additional cell type to CHA-9.3 cells. Thus, we used $p53^{-/-}$ mouse embryo fibroblasts (MEF). We treated these MEFs either with the AKT inhibitor MK-2206 or with ETP-47037 at 10 μM concentration for 24 h and analyzed TRF1 foci by immunofluorescence. The results clearly show that both the AKT inhibitor and our ETP-47037 proprietary compound induce a 22% and 34% reduction in TRF1 foci intensity, respectively (Fig. 3a).

Together, these results demonstrate that the PI3K(p110α)-AKT pathway is involved in TRF1 regulation, while the downstream kinase mTOR is not necessary for this inhibition.

**Depletion of the p110α isoform of PI3K reduces TRF1 foci.** Next, we used a genetic approach to validate the above-described chemical data indicating that the PI3K p110α isoform is the one responsible for inhibition of TRF1 foci. To this end, we used the Cre-LoxP mediated recombination system to conditionally inactivate either the *PIK3CA* or *PIK3CB* genes encoding the p110α and p110β catalytic subunits, respectively, in the corresponding immortalized p110α (lox/lox) and p110β (lox/lox) MEFs[75] ("Methods" section). We first infected the conditional knockout MEFs with adenovirus simultaneously expressing Cre recombinase and GFP (Adeno-Cre-GFP). In this manner, the GFP-positive cells will also express Cre, thus marking those cells that carry each *PIK3C* gene deletion. We next quantified TRF1 foci by immunofluorescence in both GFP-positive and GFP-negative cells (Fig. 3b, c). In agreement with the chemical inhibitor data (Figs 2d and 3a), genetic ablation of the gene encoding the p110α catalytic subunit but not of the one encoding p110β catalytic subunit strongly inhibited TRF1 foci intensity (Fig. 3b, c; note that eGFP-positive cells deleted for p110α but not for p110β show dramatically decreased TRF1 foci in the representative images). In summary, both chemical and genetic inhibition approaches demonstrate that formation of TRF1 foci is regulated by PI3Kα but not by the PI3Kβ isoform.

**PI3K-AKT inhibition leads to telomere fragility.** *Trf1* abrogation is known to lead to a persistent DDR at telomeres, and to induction of telomere-induced chromosomal aberrations[28–30]. *Trf1* genetic deletion induces the so-called multitelomeric signals (MTS), a chromosomal aberration related to increased telomere fragility during DNA replication[28, 29]. In agreement with their activity against TRF1 foci, we previously demonstrated that the ETP-47037 and ETP-47228 inhibitors can also induce formation of MTS in IPSCs (IPSC)[39]. Here, we set out to demonstrate a direct role of PI3K inhibition in the induction of MTSs provoked by PI3K inhibitors. To this end, we treated CHA 9-3 cells with either DMSO, the AKT inhibitor MK-2206 (AKTi) and the PI3K inhibitors ETP-47037 or ETP-47228. For detection of MTS, we performed quantitative fluorescence in situ hybridization (Q-FISH) analysis on metaphase spreads using a telomeric probe to visualize telomeres ("Methods" section). MTS are visualized by presenting a multi-dot pattern at chromosome ends in contrast to normal telomeres that show a single dot at each chromosome end. In agreement with their activity against TRF1 foci, treatment with ETP-47037, ETP-47228 and with AKTi significantly increased the frequency of MTS events per metaphase compared to DMSO-treated cells (Fig. 3d, e).

We next set to genetically validate these findings using immortalized $p110\alpha^{lox/lox}$ MEFs transduced with Cre recombinase to induced *P110α* deletion. MTS were determined in

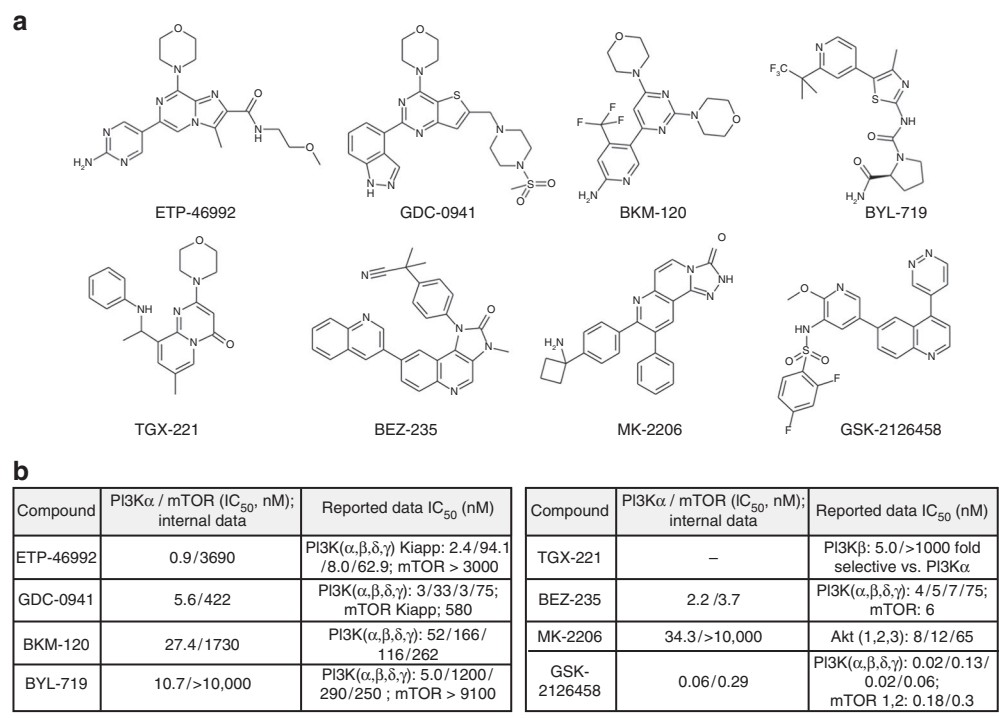

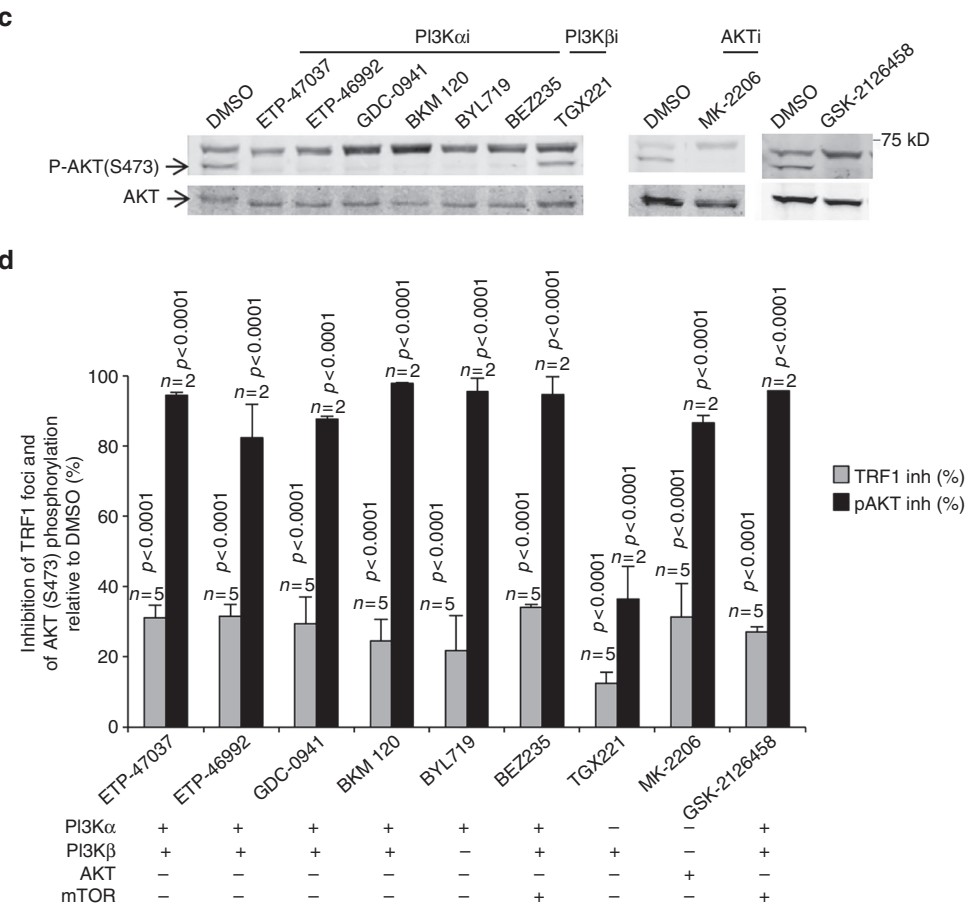

**Fig. 2** TRF1 regulation by PI3K and AKT inhibitors. **a** Structurally diverse PI3K and PI3K/mTOR inhibitors used in the study. **b** PI3K and mTOR data generated internally and reported in literature. **c** Representative western blot images of phosphorylated AKT-Ser473 and total AKT in CHA-9.3 mouse lung tumor cell line at 24 h after treatment with PI3K, AKT, and mTOR inhibitors as indicated. **d** Percent inhibition of TRF1 foci by immunofluorescence and of AKT phosphorylation at S473 (pAKT) in CHA-9.3 mouse lung tumor cell line at 24 h after treatment with the indicated inhibitors relative to TRF1 levels and to pAKT levels in control cells treated with DMSO. The inhibitors were tested at 10 μM except GSK-2126458 that was used at 1.0 μM (in **c** and **d**). Error bars represent standard deviation. The Student's *t* test was used for statistical analysis; *P* values are shown. *n* number of independent experiments

chromosome spreads as described above. Conditional $p110\alpha^{lox/lox}$ MEFs were transduced with either Cre or GFP. In agreement with the results shown with PI3K and AKT chemical inhibitors, p110α deletion induced a significant threefold increase in the incidence of MTS compared to GFP transduced cells (Fig. 3f).

**The PI3K-AKT pathway regulates TRF1 protein levels.** We investigated whether TRF1 inhibition by PI3K and AKT inhibitors was occurring at the level of TRF1 transcription or at the post-transcriptional level. To this end, CHA 9-3 lung cancer cells were incubated with 10 μM of PI3K inhibitors (ETP-47037 and GDC-0941) or AKT inhibitor (MK-2206) for a total of 24 h. None of the inhibitors significantly affected TRF1 mRNA levels (Supplementary Fig. 2a). As genetic validation, we also found normal *Trf1* mRNA levels in MEFs genetically deleted for *P110α*, the gene encoding PI3Kα (Supplementary Fig. 2b), thus indicating that regulation of TRF1 foci by PI3K/AKT does not occur at the transcriptional level. Next, we performed western blot analysis to determine whether PI3K/AKT could regulate TRF1 protein levels (Fig. 4a). Interestingly, chemical inhibition of either AKT (using the MK-2206 AKTi) or PI3K (using ETP-47037 and ETP47228)

lead to significantly decreased TRF1 protein levels in nuclear extracts (Fig. 4a).

We confirmed these findings using $p53^{-/-}$ MEFs as an independent cell type. Treatment of these MEFs with the PI3K ETP-47037 or with the AKT inhibitor MK-2206 also showed decreased total nuclear TRF1 levels compared to cells treated with DMSO (Fig. 4b).

These findings were further validated genetically using immortalized $p110\alpha^{lox/lox}$ MEFs. $P110\alpha^{lox/lox}$ MEFs transduced with Cre recombinase showed decreased total nuclear TRF1 foci fluorescence compared to non-transduced cells (Fig. 4c).

Together, these findings indicate that PI3K/AKT inhibitors decrease TRF1 protein levels. We next explored whether inhibition of the proteasome by bortezomib could rescue TRF1 protein levels. Bortezomib treatment lead to a twofold increase in TRF1 intensity levels in CHA 9-3 cells (Fig. 4d). However, combined treatment of CHA 9-3 cells with bortezomib and PI3K inhibitors ETP-47037, ETP-47228, or with the AKT inhibitor MK-2206 resulted in significantly lower TRF1 levels as compared to bortezomib/DMSO-treated control cells suggesting that PI3K/AKT-mediated TRF1 regulation is proteasome-independent (Fig. 4d).

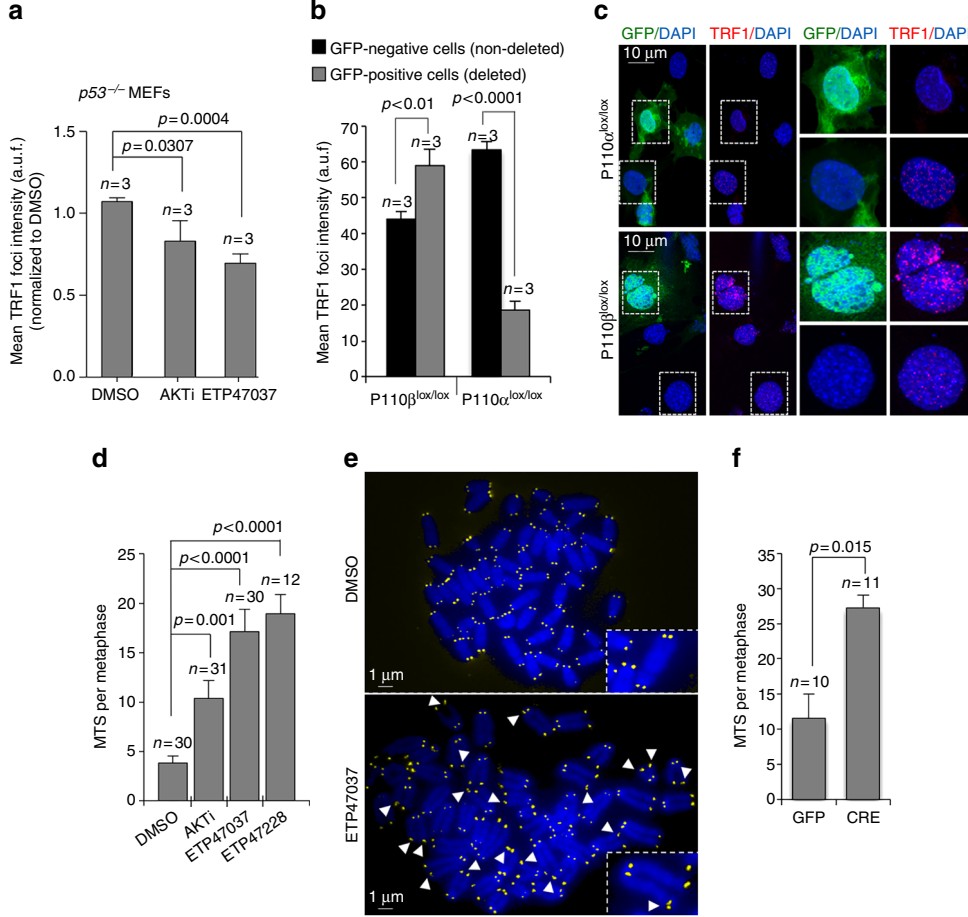

**Fig. 3** TRF1 telomeric levels are regulated by p110α. **a** Quantification of mean TRF1 foci intensity by immunofluorescence of immortalized $p53^{-/-}$ MEFs were treated either with DMSO, AKT inhibitor (AKTi), or with ETP47037. **b** Quantification of mean TRF1 foci intensity by immunofluorescence in p110α (lox/lox) and p110β (lox/lox) MEFs infected with Adeno-CRE-GFP. The graph shows quantification of TRF1 foci in both infected (Ad-CRE-GFP) and non-infected (negative) cells. **c** Representative TRF1 (red) and GFP (green) immunofluorescence images. Scale bars are shown. **d** Incidence of multitelomeric signals (MTS) events in metaphase spreads of lung cancer-derived cells (CHA 9-3) treated either with DMSO, AKT inhibitor (AKTi) (10 μM), ETP47037 (10 μM), or ETP47228 (10 μM). **e** Representative images of metaphase spread corresponding to DMSO- and to ETP47037-treated cells. MTS are labeled with a white arrow head. Zoomed images are shown in the insets. Scale bars are shown. **f** Incidence of multitelomeric signals (MTS) events in metaphase spreads of immortalized $P110\alpha^{lox/lox}$ MEFs either transduced with GFP or with Cre recombinase. (*n*) Analyzed metaphases in each condition. The Student's *t* test was used for statistical analysis; *P* values are shown. Error bars represent standard deviation. *n* number of independent experiments

ADP-ribosylation of human TRF1 by tankyrase 1 releases TRF1 from telomeres, which serves as a signal for its ubiquitination and subsequent degradation[16, 19]. Although the mouse TRF1 lacks the canonical Tankyrase 1 binding site and has been previously reported not to be a substrate for tankyrase 1 poly (ADP-rybosyl)sylation in vitro[76], here we nevertheless tested whether a Tankyrase inhibitor was able to abrogate the effects induced by PI3K inhibitors. However, in agreement with previous reports, a tankyrase inhibitor did not rescue TRF1 down-regulation by PI3K inhibitors (ETP-47037) (Supplementary Fig. 3).

**Purified TRF1 is directly phosphorylated by AKT.** To address whether TRF1 is a direct substrate of AKT, affinity purified GST-TRF1 was incubated with purified human AKT1 (1379-0000-2, ProQinase) in the presence of γ-$^{32}$ATP. As positive control, a GSK3 (14-27) peptide (0349-0000-5, prokinase) was also used as substrate. AKT1 yielded a TRF1 phosphorylation signal after incubation with purified GST-TRF1 during 1 h, which was decreased in the presence of AKT inhibitor (see asterisk in Fig. 5a,

left). AKT1 also yielded a phosphorylation signal in the positive control peptide GSK3 (14-27) that was reduced in the presence of AKT inhibitor (Fig. 5a, right). An autophospholyration activity of AKT1 was detected after 1 h of incubation and this activity was reduced in the presence of AKT inhibitor (Fig. 5a). Together, these findings indicate that TRF1 is a direct substrate of AKT1.

We next set to identify the specific TRF1 phosphorylation sites by AKT. To this end, we analyzed phosphopeptides by liquid chromatography-mass spectrometry (LC/MS/MS) analysis in TRF1 and TRF1&AKT1, samples containing ATP ("Methods" section). We identified two different TRF1 phosphopeptides in samples containing AKT1. One peptide presented threonine 330 (T330) and the other contained phosphorylated threonine 248 (T248) (Fig. 5b–e). As expected, no phosphopeptides were identified in the sample containing only TRF1 (Fig. 5d, e). Of note, human TRF1 has been shown to be phosphorylated by AKT at residue T273, which corresponds to residue S344 in mouse TRF1[24]. However, this potential TRF1 phosphorylation site in residue S344 was not identified in our LC/MS/MS analysis, most likely due to the fact that this site is located between two close arginine residues, thus, trypsin digestion renders a 4-amino acid

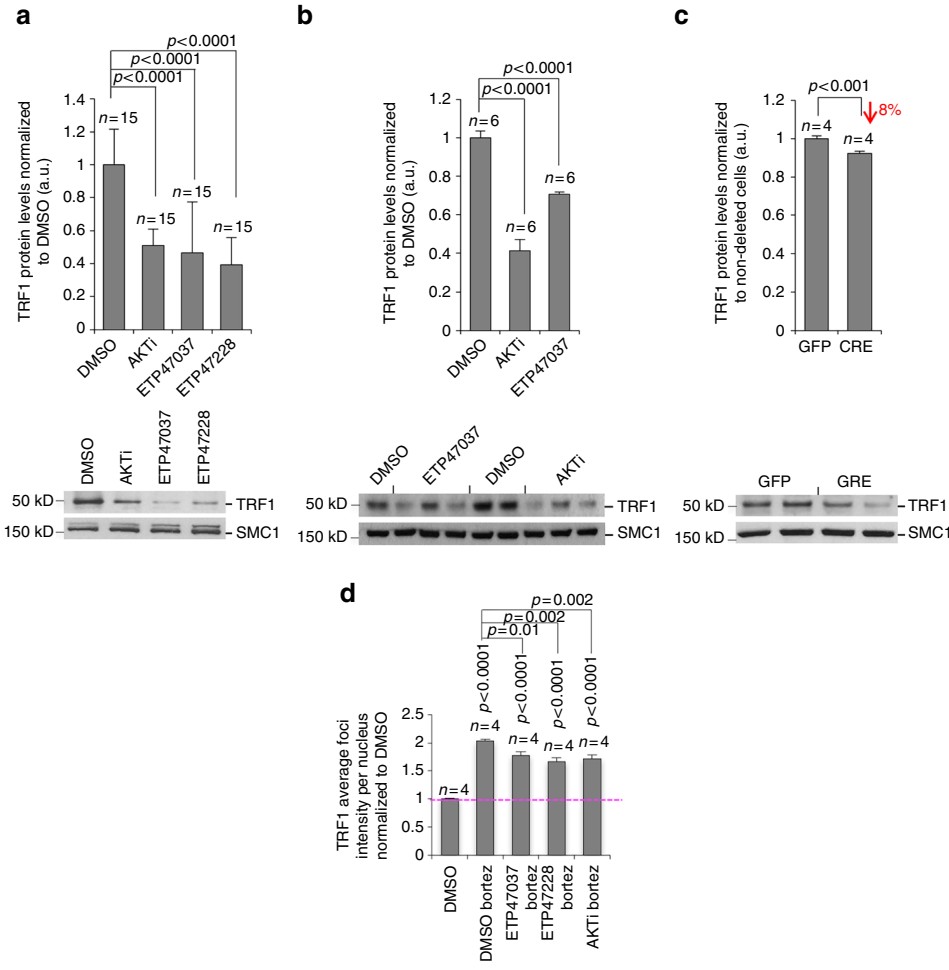

**Fig. 4** PI3K/AKT inhibitor post-translationally downregulate TRF1 levels. **a** Quantification of total nuclear TRF1 levels in lung cancer-derived cells (CHA 9-3) treated either with DMSO, AKT inhibitor (AKTi) (10 μM), ETP47037 (10 μM), or ETP47228 (10 μM). A representative western blot image is shown below. **b** Quantification of total nuclear TRF1 levels in in immortalized p53$^{-/-}$ MEFs treated either with DMSO, AKT inhibitor (AKTi) (10 μM), or ETP47037 (10 μM). A representative western blot image is shown below. **c** Quantification of total nuclear TRF1 levels in P110α$^{lox/lox}$ MEFs cells transduced with pBabe-GFP or with pBabe-Cre recombinase. A representative WB images is shown below. **d** Quantification by immunofluorescence of mean TRF1 foci intensity in CHA 9-3 cells treated either with DMSO, DMSO plus bortezomib (50 nM), ETP47037 (10 μM) plus bortezomib (50 nM), ETP47228 (10 μM) plus bortezomib (50 nM), and AKTi (10 μM) plus bortezomib (50 nM). Student's t test was used for statistical analysis; P values are shown. Error bars represent standard error. n number of independent experiments

peptide (TSGR) too short to be detected by conventional LC/MS/MS.

In order to in vitro validate these phosphorylation sites including the S344 residue, we generated GST-tagged *Trf1* alleles where either the T248, T330, or S344 residues were mutated to alanine. We also generated a *GST-Trf1* allele that contained the triple substitution T248A/T330A/S344A (Fig. 5f). Affinity

purified GST-TRF1, GST-TRF1$^{T330A}$, GST-TRF1$^{S344A}$, GST-TRF1$^{T248A}$, and GST-TRF1$^{T330A/S344A/T248A}$ were incubated with purified human AKT1 in the presence of γ-$^{32}$ATP (Fig. 5g). We found significantly decreased levels of phosphorylated-TRF1 in the variants harboring T330A and S344A substitutions compared to wild-type TRF1 (Fig. 5g). The T248A substitution also rendered approximately a 15% decreased in TRF1

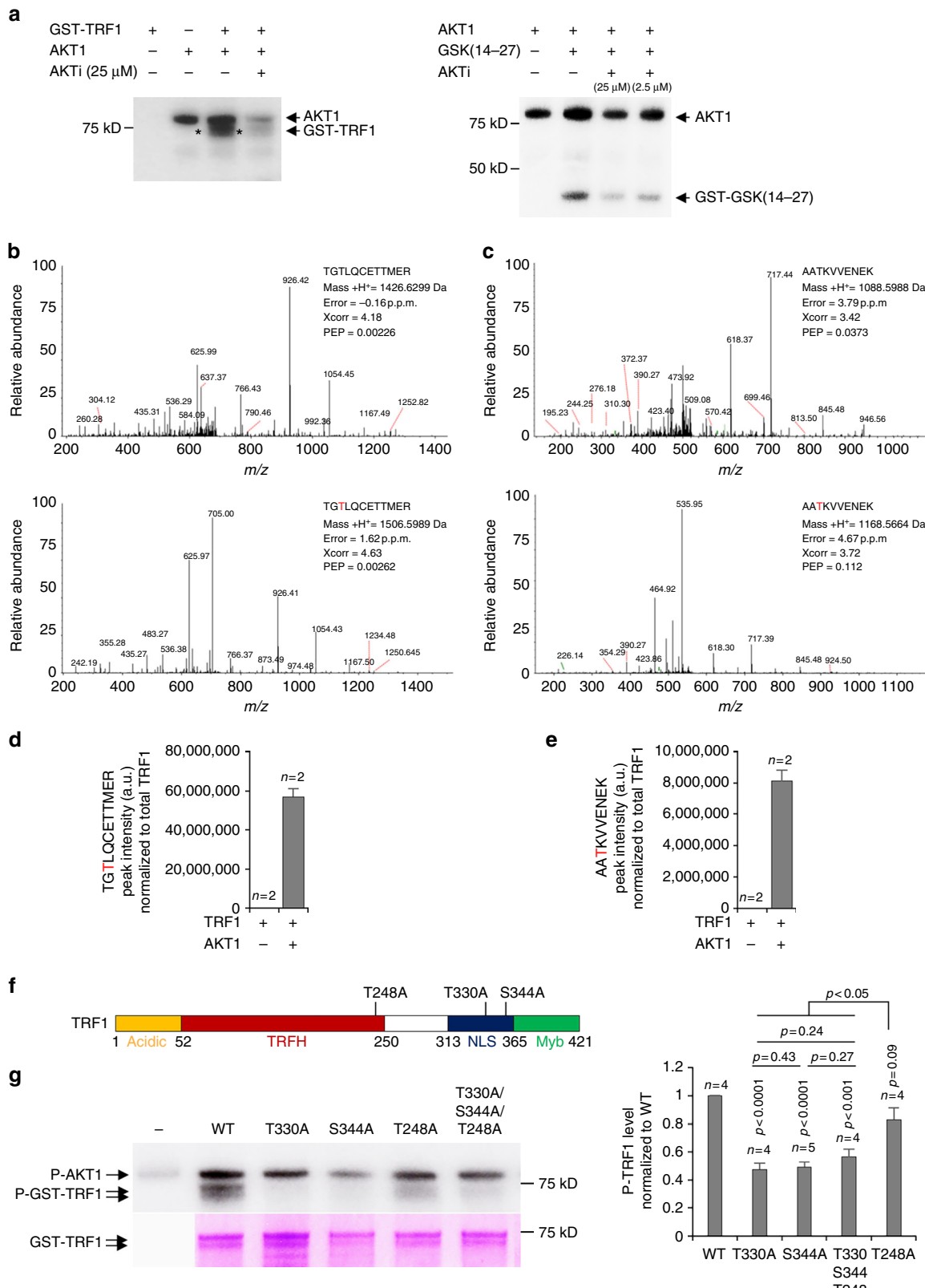

phosphorylation levels by AKT1 although the difference did not reach statistical significance (Fig. 5g). In summary, AKT1-dependent phosphorylation of TRF1 is reduced by mutation of T330 and S344 residues and to a lesser extent by mutation of the T248 residue. Of note, the observation that there is a detectable phosphorylated-TRF1 signal in the absence of the identified AKT phosphorylation sites (triple mutant TRF1$^{T330A/S344A/T248A}$) may indicate the presence of additional sites within TRF1.

**TRF1 phosphorylation by AKT required for TRF1 foci formation.** Finally, to address the *in vivo* role of these TRF1 modifications by AKT1, we transduced *eGFP*-tagged *Trf1* wild-type and mutant alleles in both primary p53-deficient MEFs (*Trf1*$^{T248A}$, *Trf1*$^{T330A}$, *Trf1*$^{S344A}$, *Trf1*$^{T330A/S344A}$, and *Trf1*$^{T248A/T330A/S344A}$ mutants) and in CHA 9-3 lung cancer cells (*Trf1*$^{T330A}$, *Trf1*$^{S344A}$, and *Trf1*$^{T330A/S344A}$ mutants) (Fig. 6a). The mean GFP spot fluorescence intensity per cell was quantified by using Opera high-content screening (HCS) (Fig. 6a). In MEFs, we found that the TRF1 single mutants T330A and S344A, as well as the TRF1 double and triple mutants (T330A/S344A and T248A/T330A/S344A) showed a significant decrease in the intensity of TRF1 telomeric foci fluorescence compared to MEFs expressing wild-type TRF1 (Fig. 6b, d). Similarly, in CHA 9-3 cells, the intensity of TRF1 telomeric foci fluorescence was significantly decreased in cells expressing the TRF1 mutants T330A, S344A, and T330/S344A, thus confirming the results obtained in MEFs (Fig. 6c, d).

In order to rule out potential interference of the GFP-tagged TRF1 variants with the endogenous TRF1, we transduced *eGFP*-tagged *Trf1* wild-type and mutant alleles in *p53*$^{−/−}$ *Trf1*$^{lox/lox}$ MEFs (*Trf1*$^{T248A}$, *Trf1*$^{T330A}$, *Trf1*$^{S344A}$, and *Trf1*$^{T330A/S344A}$ mutants) and then deleted the endogenous *Trf1* by transduction with the Cre recombinase. Overexpression of *eGFP-Trf1* alleles and complete deletion of the endogenous *Trf1* was confirmed by western blot analysis using a specific TRF1 antibody (Fig. 6e). Quantification of mean GFP spot fluorescence intensity in *Trf1*$^{−/−}$ MEFs confirmed that the TRF1 single mutants T330A and S344A, as well as the TRF1 double mutant (T330A/S344A) showed a significant decrease in the intensity of TRF1 telomeric foci compared to MEFs expressing wild-type TRF1 (Fig. 6f). In contrast, no significant differences were detected between TRF1-T248A and wild-type TRF1 (Fig. 6f). In order to address the ability of these mutant variants to rescue the severe proliferative defects associated to TRF1 deficiency[28, 29], we analyzed growth rate in *Trf1*-deficient MEFs expressing *eGFP*-tagged *Trf1*$^{wt}$, *Trf1*$^{T248A}$, *Trf1*$^{T330A}$, and *Trf1*$^{S344A}$ alleles (Fig. 6g). We found that cells expressing TRF1-T330A and TRF1-S344A did not rescue impaired proliferation in *Trf1*-deleted cells, indicating that these mutant TRF1 proteins are non-functional in their ability to complement proliferation defects in TRF1-deleted cells. In

contrast, cells expressing TRF1-T248A grew at similar rate than TRF1 wild-type cells, suggesting that TRF1-T248A variant has no major impact on TRF1 function (Fig. 6g).

These observations are in agreement with the in vitro phosphorylation data of TRF1 by AKT1 in which T330A, S344A, and T248A/T330A/S344A mutants rendered reduced P-TRF1 levels (Fig. 5g). These results indicate that AKT1 specifically phosphorylates TRF1 at T248, T330, and at S344, and that mutation of T330 and S344 but not of T248 sites to non-phosphorylatable alanine residues impairs proper TRF1 function.

**AKT regulates TRF1 half-life and binding to telomeres.** Decreased AKT-mediated TRF1 phosphorylation results reduction of total TRF1 protein amounts (Fig. 4) and in decreased of TRF1 foci (Figs 1d, 2d, 3a, b and 6b–e). We envisioned at least two scenarios in which AKT-mediated TRF1 phosphorylation could be regulating TRF1 levels at telomeres. On the one hand, AKT-mediated phosphorylation of TRF1 may regulate TRF1 stability and turn-over. On the other hand, AKT-mediated phosphorylation of TRF1 might also regulate TRF1 binding to telomeres, either by affecting TRF1 dimerization and/or TRF1 telomeric DNA-binding capacity[77]. To address the mechanisms underlying PI3K/AKT-dependent TRF1 regulation, we first analyzed the half-life of the GFP-tagged wild-type and non-phosphorylatable T248A, T330A, S344A, and T330A/S344A TRF1 variants. The results showed that all the mutant proteins present a twofold reduction in their half-life (HL <2.5 h) as compared to wild-type TRF1 (HL = 4.5 h), indicating that AKT-mediated phosphorylation is involved in TRF1 stability (Fig. 7a).

To address the dimerization capability of the non-phosphorylatable TRF1 mutants, we performed western blot analysis of nuclear cell extract of *Trf1*-deleted MEFs expressing GFP-tagged wild-type and T248A, T330A, S344A, and T330A/S344A TRF1 variants in non-reducing conditions (Fig. 7b). While in reducing conditions GFP-TRF1 renders a 70 KD band, in non-reducing conditions without DTT a band of approximately 140 kD was detected in GFP-TRF1 wild-type and all the mutant variants analyzed, corresponding to TRF1 dimers. The results clearly show that none of these mutations affect the dimerization capacity of TRF1 (Fig. 7b).

We next sought to determine whether the non-phosphorylatable TRF1 mutant proteins were affected in their ability to bind telomeric DNA. To this end, affinity purified GST-TRF1, GST-TRF1$^{T330A}$, GST-TRF1$^{S344A}$, and GST-TRF1$^{T248A}$ were incubated with radiolabeled telomeric DNA and with TRF1 specific antibody. Visualization of the protein-DNA complexes by electrophoretic mobility shift assay (EMSA) confirmed that wild-type TRF1 and TRF1$^{T248A}$ efficiently bound telomeric DNA, whereas GST-TRF1$^{T330A}$ and GST-TRF1$^{S344A}$ bound telomeric DNA less efficiently (Fig. 7c), in agreement with

**Fig. 5** TRF1 is phosphorylated in vitro by AKT1. **a** One μg of purified GST-TRF1 was incubated with 0.2 μg of human GST-AKT1 with or without AKT inhibitor (MK-2206) in kinase buffer containing 5 μCi [γ-$^{32}$-P]ATP (left). The peptide RBER-GSK3(14-27) was used as positive controls (right). The reaction mixtures were resolved by SDS-PAGE and subjected to autoradiography. **b** MS/MS spectrum of unmodified TRF1 328-339 (above panel) and of the TRF1 328-339 peptide in which phosphorylation was assigned to T330 residue (below panel). **c** MS/MS spectrum of unmodified TRF1 246-255 peptide and of the TRF1 246-255 peptide in which phosphorylation was unambiguously assigned to the T248 residue (below panel). Mass error, identification score, and posterior error probability (PEP) are shown for the different peptides. **d**, **e** Quantification of TGTLQCETTMER (T330) (**d**) and of the AATKVVENEK (T248) (**e**). Phosphopeptide peak intensity normalized to total TRF1 signal in samples containing only TRF1 or TRF1 plus AKT1. **f** Schematic representation of TRF1 protein depicting its N-terminal acidic domain, TRFH domain, nuclear localization signal (NLS), and its Myb domain. The residues found phosphorylated by AKT (T248, T330, and S344) were mutated to alanine. **g** Representative image of in vitro phosphorylation assay of purified GST-TRF1, GST-TRF1(T330A), GST-TRF1 (S344A), GST-TRF1 (T248A), or GST-TRF1 (T330A/S344A/T248A) by GST-AKT1. The reaction mixtures were resolved by SDS-PAGE and subjected to autoradiography. As loading control, purified GST-TRF1 protein samples were run in SDS-PAGE and coomassie-stained (below panel). The quantification of mutant P-TRF1 levels normalized to wild-type TRF1 is shown to the right. Student's *t* test was used for statistical analysis, *P* values are shown. Error bars represent standard deviation. *n* number of independent experiments

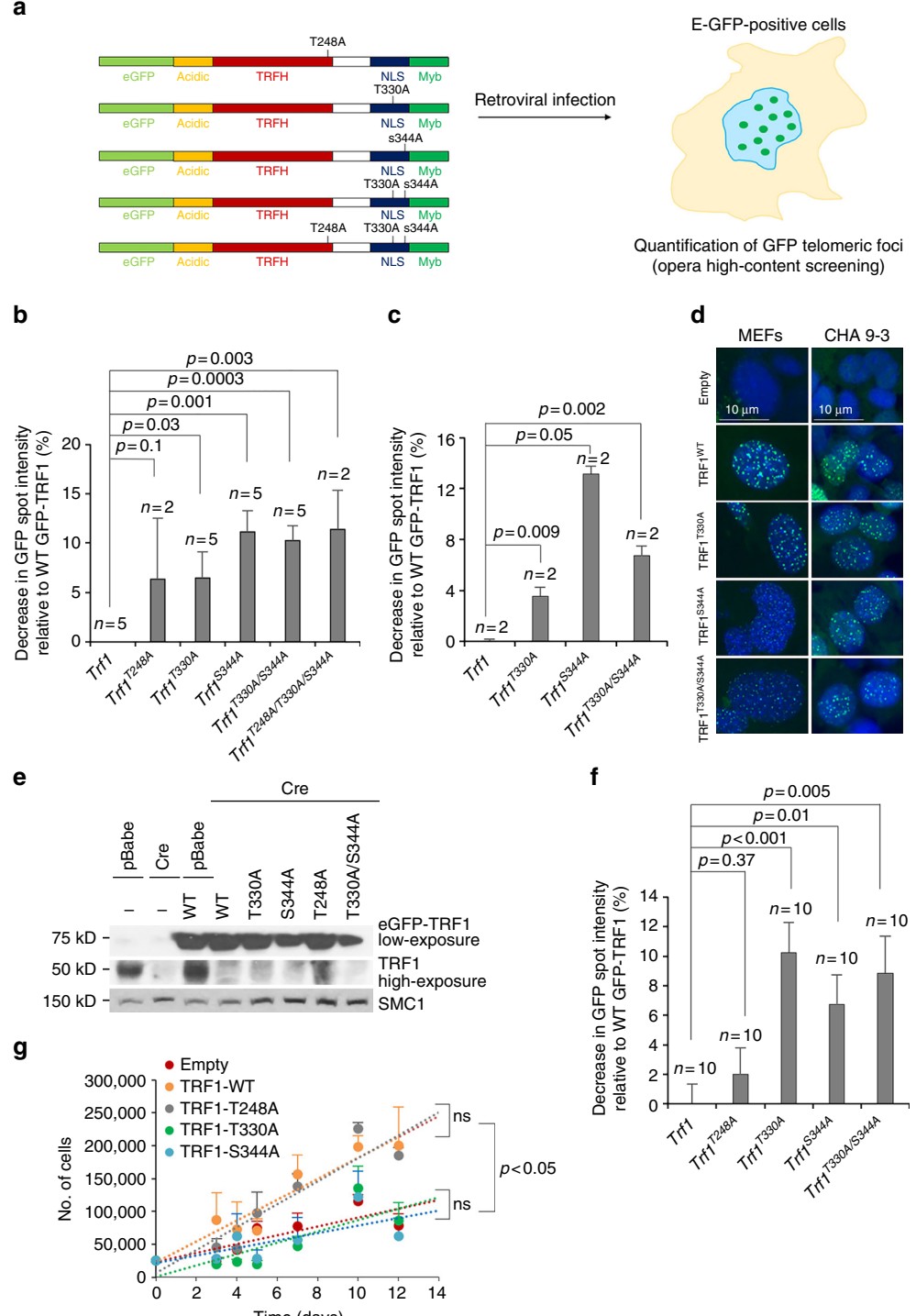

**Fig. 6** Phosphorylation of T248, T330, and S344 in TRF1 stabilizes TRF1 telomeric foci. **a** Retroviral infection of *eGFP*-tagged *Trf1* WT or mutant alleles (*Trf1^T248A^*, *Trf1^T330A^*, *Trf1^S344A^*, *Trf1^T330A/S344A^*, and *Trf1^T248A/T330A/S344A^*) was performed to overexpressed depicted TRF1 variants. The mean GFP spot intensity per cell was quantified by Opera high-content screening (HCS). **b**, **c** Quantification of the decrease in mean GFP-TRF1 spot telomeric intensity in transduced wild-type MEFs (**b**) and in CHA 9-3 cells (**c**). The percent change in mean intensity of TRF1 mutant variants was represented relative to wild-type TRF1. **d** Representative eGFP-TRF1 immunofluorescence images. Scale bars are shown. **e** *Trf1^lox/lox^* MEFS with or without overexpression of *eGFP-Trf1* WT or mutant (*Trf1^T248A^*, *Trf1^T330A^*, *Trf1^S344A^*, and *Trf1^T330A/S344A^*) alleles were transduced with the Cre recombinase to delete endogenous *Trf1*. As controls, *Trf1^lox/lox^* and *Trf1^lox/lox^* overexpressing WT eGFP-TRF1 were infected with pBabe. Expression of the eGFP-TRF1 variants and of endogenous TRF1 was determined by western blot using a specific TRF1 antibody. **f** Quantification of the decrease in mean GFP-TRF1 spot telomeric intensity in transduced *Trf1^Δ/Δ^* deleted MEFs. **g** Growth rate of *Trf1^Δ/Δ^* deleted MEFs overexpressing *eGFP*-tagged *Trf1* WT and mutant alleles as indicated. Student's *t* test was used for statistical analysis and *P* values are shown. Error bars represent standard errors. *n* number of independent experiments

the in vitro TRF1 phosphorylation data (Fig. 5f, g) and the in vivo TRF1 foci formation and cell proliferation data (Fig. 6a–g). Specificity of the protein-DNA complexes was demonstrated by the supershift observed upon addition of TRF1 antibody to the reaction (Fig. 7c).

These results indicate that AKT1-mediated phosphorylation of TRF1 at T330 and at S344 regulate TRF1 protein stability and are

required for proper TRF1 telomeric DNA binding and cell viability.

**Decreased TRF1 levels in PDXs treated with PI3Kα inhibitor.** We next set to address whether PI3Kα inhibition in human tumor samples showed decreased TRF1 levels compared to untreated controls. To this end, we generated PDXs from seven independent

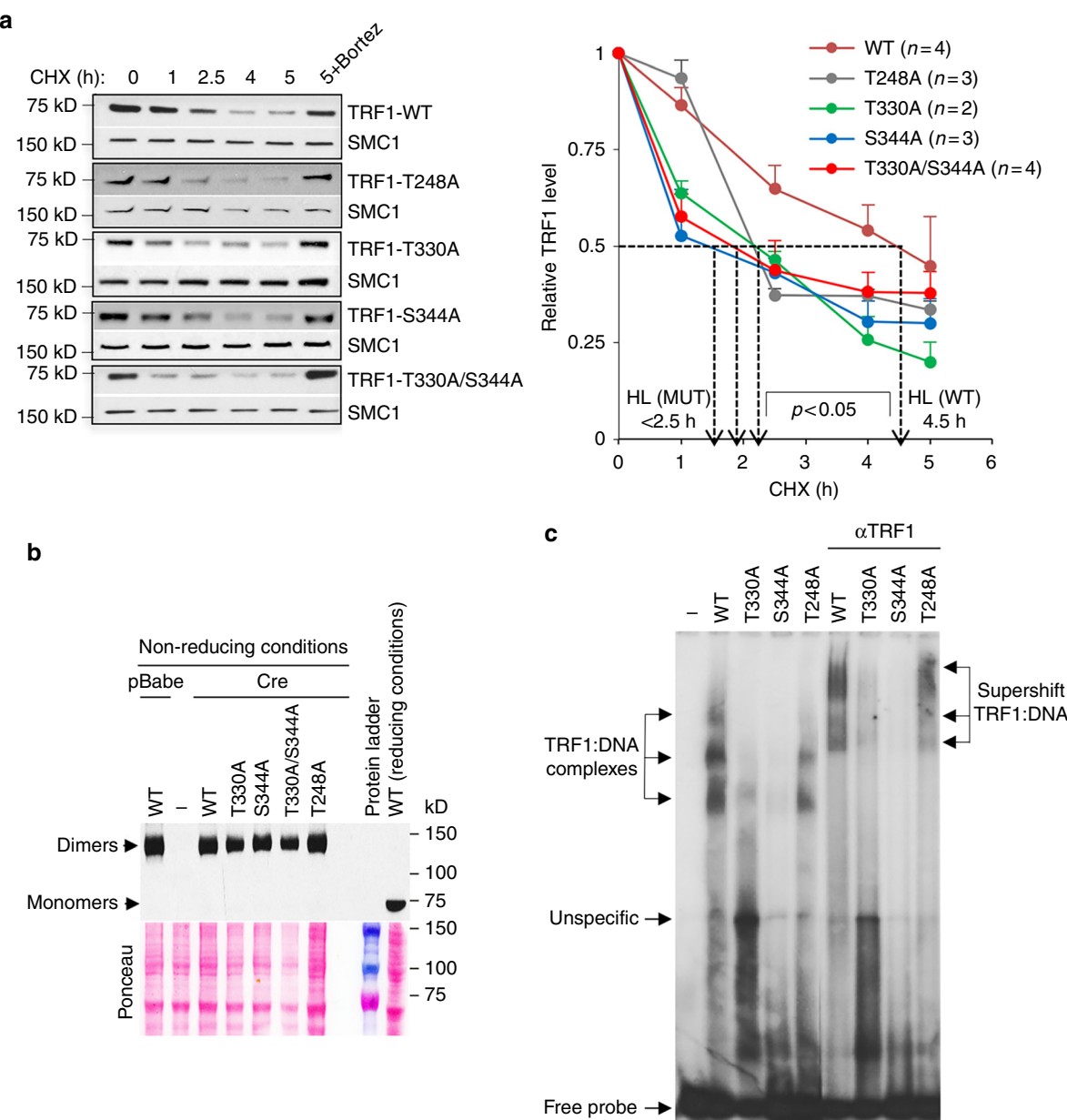

**Fig. 7** AKT-mediated TRF1 phosphorylation regulate TRF1 protein stability and telomeric DNA-binding ability. **a** TRF1 protein half-life (HL) was determined in MEFs overexpressing $eGFP$-$Trf1$ WT or mutant ($Trf1^{T248A}$, $Trf1^{T330A}$, $Trf1^{S344A}$, and $Trf1^{T330A/S344A}$). Cells were pre-treated with 100 μg/ml cyclohexamide (CHX) or bortezomib (5 μM) for 30 min. At the indicated times, the cells were collected to prepare nuclear extracts, followed by western blot analysis against TRF1 or SMC1 (as loading control) (left). The graph represents the mean value for each time point and corresponding standard deviation (right panel). $P$ values are indicated. **b** $Trf1^{lox/lox}$ MEFS with or without overexpression of $eGFP$-$Trf1$ WT or mutant ($Trf1^{T248A}$, $Trf1^{T330A}$, $Trf1^{S344A}$, and $Trf1^{T330A/S344A}$) alleles were transduced with the Cre recombinase to delete endogenous $Trf1$. As control, $Trf1^{lox/lox}$ MEFs overexpressing WT eGFP-TRF1 was infected with pBabe. Nuclear protein extracts were prepared in non-reducing conditions (without DTT) to detected TRF1 dimers. Western blot against TRF1 was performed in non-reducing conditions without heating in SDS-PAGE. A single sample corresponding to WT-TRF1 was loaded in reducing conditions to show the monomeric form of TRF1. Ponceau staining is shown as loading control (below panel). **c** In vitro TRF1 binding to telomeric dsDNA (TTAGGG)$_7$ assay was performed with GST-tagged TRF1 WT and mutant variants (T330A; S344A and T248A) as indicated by electrophoretic mobility shift assay (EMSA). Arrows show the positions of the three TRF1-containing complexes observed. As control for specificity of TRF1/DNA interactions, specific TRF1 antibody was also added to the reaction to obtain a supershift

ER/Her2-positive breast cancer patients in female athymic NMRI nu/nu mice, and treated these with either the PI3Kα specific inhibitor BYL719 (35 mg/k 6IW) or with vehicle during a total of 12 days. Mice were then killed and PDXs extracted and formalin fixed for further analysis. A TRF1 immunofluorescence followed by a telomere quantitative q-FISH was performed in PDXs sections to identify TRF1 specifically located at telomeres by co-localization of TRF1 protein to telomeric DNA, as well as to quantify telomere length. Telomeric TRF1 foci at telomeres (green spots) were unequivocally identified by co-localization with telomeric PNA probe (red spots) (Fig. 8a). Quantification of telomeric TRF1 spot intensity by immunofluorescence in seven independent PDXs (in duplicates) showed that TRF1 levels significantly decreased in 4 out of 7 PDX (PDX 153, PDX131, PDX313, and PDX244), increased in 2 out of 7 PDX (PDX39 and PDX191), and remained unchanged in one PDX (PDX98) (Fig. 8a, b). Immunohistochemistry staining of phosphorylated AKT (S473) in these same PDXs followed by quantification of tumor positive area was performed to address the efficacy of BYL719 treatment (Fig. 8c, d). In 4 out of 7 PDXs (PDX153, PDX313, PDX98, and PDX244) BYL719 treatment induced a decrease in pAKT (S473) as compared to vehicle treated controls, indicating efficacy in PI3K pathway inhibition. These four PDXs were considered as "responders" to the PI3Kαi treatment. Two PDXs, PDX131 and PDX39, were negative for pAKT at basal levels (before treatment with the PI3Ki BYL719) and remained unchanged upon BYL719 treatment. Interestingly, PDX191 showed a significant 46% increase in pAKT levels and 80% increase in telomeric TRF1 levels in treated samples as compared to untreated controls. This unexpected increase in pAKT levels of PDX191 upon BYL719 treatment could be explained by the activation of regulatory feedback loops that trigger compensatory signaling events leading to activation of the AKT pathway[78]. These three PDX (PDX131, PDX39, PDX191) were considered as "non-responders" to the treatment with the PI3Kα inhibitor BYL719. Of note, in 3 of the 4 responders (PDX153, PDX313, and PDX244) the telomeric TRF1 levels significantly decreased after BYL719 treatment while in the other PDX (PDX98) the TRF1 levels remained the same. It should be pointed out that untreated PDX131 and PDX39 that presented undetectable pAKT levels correspond to those samples with the lowest telomeric TRF1 levels (Fig. 8b, d). Quantification of mean telomere length in these PDX samples showed moderate but significant changes in mean telomere length upon BYL719 treatment. Thus, TRF1 levels correlated significantly both with telomere length and with pAKT (S473) levels, although the correlation was more significant in the case of pAKT(S473) levels (Fig. 8f). In addition, the telomere length changes were more moderate as compared to changes observed in TRF1 levels.

Decrease in pAKT (S473) levels and the concomitant decrease in TRF1 foci signal in the "responder" PDXs following treatment with the PI3Kα inhibitor BYL719 was accompanied with a significant increase in cells presenting DNA damage (γH2AX positive cells) (Fig. 8g). We also observed a higher number of telomere-induced DNA damage foci (TIFs) in the BYL719-treated "responder" tumors than in the "non-responders" (Fig. 8h, i), in agreement with lower levels of telomere bound TRF1 levels in the "responders" (Fig. 8b, d). These findings suggest that TRF1 inhibition also occurs in human tumor samples upon efficient inhibition of PI3Kα, thus pinpointing TRF1 as a relevant target of PI3K inhibitors, which leads to increase DNA damage at telomeres.

## Discussion

Telomeres are essential to maintain genome stability. TRF1 is an essential component of shelterin, the telomere protective complex. TRF1 has a fundamental role in protecting telomeres from DNA repair activities and from replication-induced fragility, thus preventing telomeric DNA damage[28, 29]. In a previous work, we identified two small molecules that disrupt TRF1 telomeric foci, however, the mechanism of action remained unknown[39]. Here, we demonstrate that these compounds regulate TRF1 location to telomeres throughout inhibition of the PI3K/AKT pathway. In particular, we show that inhibition of PI3K and AKT by using small molecules, or by genetic depletion of the gene encoding the PI3Kα subunit (p110α), leads to decreased TRF1 protein levels and decreased TRF1 telomeric foci. Furthermore, cells treated with PI3K/AKT inhibitors show increased number of telomere aberrations and telomere-induced DNA damage. Importantly, patient-derived PDX models treated with PI3K inhibitors that showed efficient downregulation of the PI3K pathway also show decreased TRF1 levels and increased telomere-induced DNA damage. These findings pinpoint TRF1, and its role in preventing telomere damage, as one of the targets of the PI3K/AKT pathway, with important implications both for cancer and aging-related therapeutic strategies.

Human AKT1 has been previously shown to interact with TRF1 and to phosphorylate TRF1 at threonine 272 by AKT[24]. Overexpression of AKT in HTC cells was also shown to induce upregulation of TRF1 protein levels and telomere shortening[24]. In contrast, here we show that lower pAKT levels before or after treatment of patient-derived PDX models of breast cancer with a PI3K/AKT inhibitor, correlates with lower TRF1 levels and shorter telomeres, although telomere length changes are more moderate compared to changes in TRF1 levels upon inhibition of the PI3K/AKT pathway. Thus, although it has been shown that long-term overexpression of TRF1 in human cells results in progressive telomere shortening[79], and TRF1 overexpression in transgenic mice also leads to telomere shortening in a XPF nuclease-dependent manner[80], here we observe longer telomeres in PDXs presenting higher endogenous TRF1 levels compared to those with lower TRF1 levels. This apparent discrepancy may be explained by the fact that in our setting TRF1 is not necessarily overexpressed, and higher levels of TRF1 in PDX with longer telomeres may reflect on the normal binding of TRF1 along longer tracts of telomeric repeats. In fact, Trf1 deletion or inhibition of TRF1 protein levels in mouse does not lead to telomere length changes but induces telomere de-protection/damage[28, 29, 39]. Nevertheless, future studies using a larger set of patient samples are needed for addressing the relationship between PI3K/AKT pathway and telomere length. Importantly, we show here a correlation between lower pAKT levels and decreased TRF1 protein localizing at telomeres in the PDX models that respond to the Pi3K/AKT inhibitors. Also, "responder" tumors show increased telomere-induced damage in agreement with telomere uncapping as the consequence of decreased TRF1 levels.

Here we demonstrate that AKT1 directly phosphorylates purified mouse TRF1 at various previously un-identified residues. In particular, by using in vitro phosphorylation assays, we identify Threonine 330 (T330) and Threonine 248 (T248) as bona fide AKT1 phosphorylation sites in mouse TRF1. In addition, we show that AKT1 also phosphorylates mouse TRF1 at Serine 344, which is the analogous site to human TRF1 residues Threonine 271 and Threonine 273, recently shown to be necessary for TRF1 binding to telomeres[24, 81]. Furthermore, we show that substitution of these residues in mouse TRF1 to non-phosphorylatable alanine residues in S344A and T330A mutants results in significantly reduced levels of in vitro AKT1-dependent TRF1 phosphorylation. An analogous trend to decreased AKT1-dependent phosphorylation was found for the T248A mutant although it did not reach statistical significance. These results demonstrate that residues S344 and T330, and to a lesser extend residue T248, in mouse TRF1 are bona fide AKT1 targets.

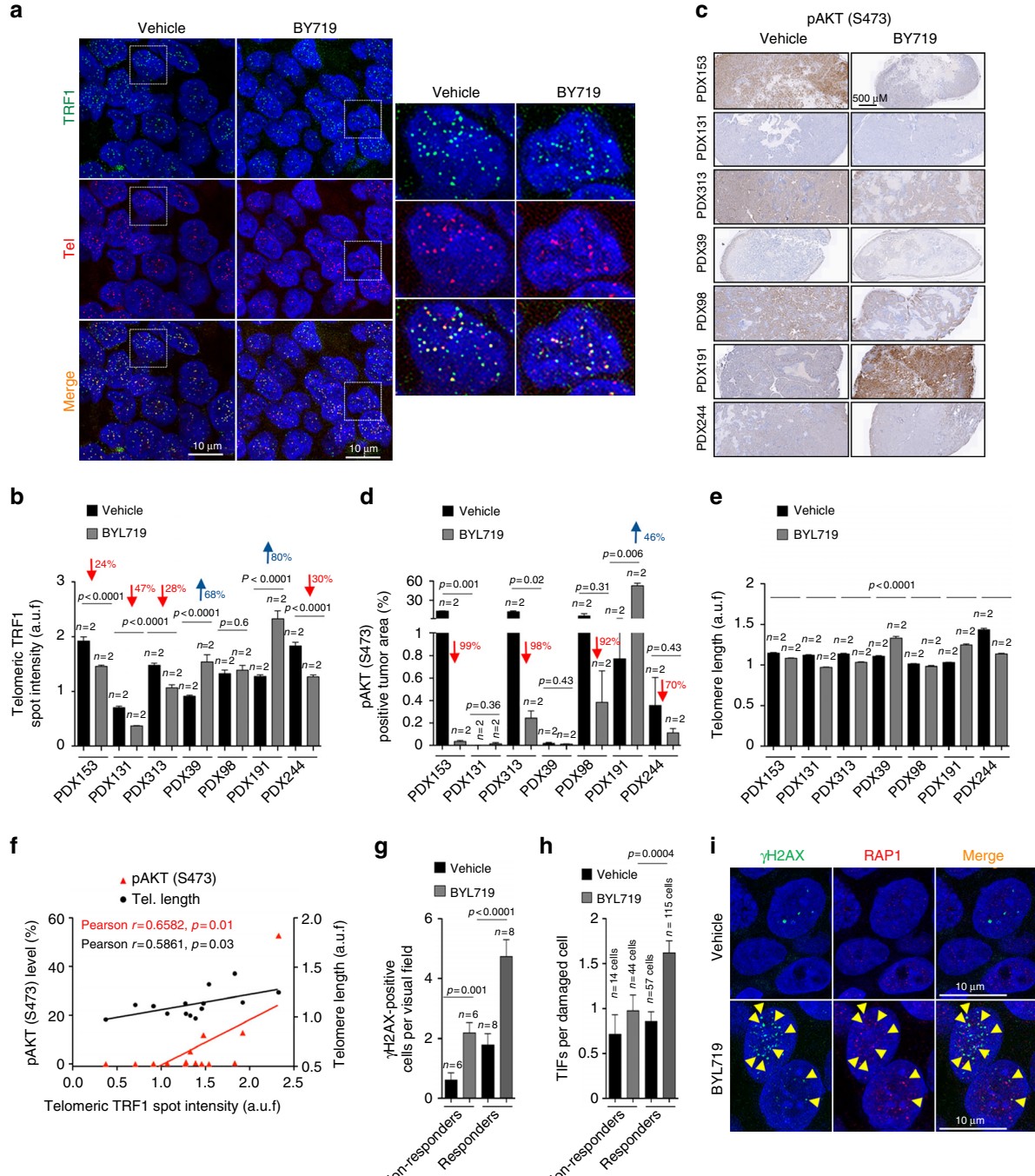

**Fig. 8** Inhibition of PI3Kα decreases TRF1 levels and increases telomeric DNA damage in human PDXs. **a** Representative immunoFish images of TRF1 immunofluorescence and telomeric DNA FISH of vehicle and BYL719-treated PDXs. Seven independent tumors were used to generate PDXs in duplicate per condition. Scale bars are shown. **b** Quantification of telomeric TRF1 foci intensity by immunofluorescence in seven independent PDXs treated either with vehicle or with BYL719 for 12 days. Telomeric TRF1 foci were identified by colocalization with telomeric DNA by FISH. The percent change in telomeric TRF1 intensity in treated PDXs as compared to non-treated are indicated in each case. Seven independent tumors were used to generate PDXs in duplicate per condition. **c** Representative immunohistochemistry images of phosphorylated AKT (S473) in PDXs treated with either vehicle or with BYL719. Scale bars are shown. **d** Quantification of pAKT (S473) positive area with regards to total tumoral area in each PDX. Scale bars are shown. The efficacy of the treatment was analyzed by quantifying the percent inhibition of pAKT (S473) as compared to vehicle treated controls and is indicated in each case. **e** Quantification of mean telomere length by Q-Fish in PDXs treated either with vehicle or with BYL719 for 12 days. **f** Pearson correlation analysis between telomeric TRF1 foci intensity and pAKT or mean telomere length of untreated and BYL179 treated PDXs. The Pearson r coefficient and the P values are indicated. **g**, **h** Number of positive γH2AX cells (≥3 foci) (**g**) and quantification of telomere-induced foci (TIFs) by double immunofluorescence with anti-RAP1 and anti-γH2AX antibodies (**h**) in PDXs treated either with vehicle or with BYL719 for 12 days. The PDXs were grouped in non-responders or responders according to the efficacy of BYL179 treatment as measured in **d**. **i** Representative RAP1 and γH2AX immunofluorescence images of vehicle and BYL719-treated PDXs. Damage cells (green) are marked with a white arrow and TIFs (yellow) with a yellow arrow head. Scale bars are shown. Student's t test was used for statistical analysis and P values are shown. Error bars represent standard errors. n number of independent experiments

Importantly, overexpression of all three eGFP-tagged TRF1 mutants in vivo, showed that T330A and S344A mutations significantly reduced telomeric eGFP-TRF1 foci in vivo, suggesting that AKT1-dependent phosphorylation of these sites is important for TRF1 binding to telomeres. Indeed, we show that these modifications regulate TRF1 stability and telomeric DNA binding. One the one hand, the half-life of non-phosphorylatable mutant proteins (T248A, T330A, and S344A) is reduced more than twofold as compared to wild-type TRF1, demonstrating that AKT-mediated phosphorylation is involved in TRF1 turn-over. On the other hand, we also show that TRF1-T330A and TRF1-S344A bound telomeric DNA less efficiently as compared to wild-type TRF1, demonstrating that these phosphorylations (T330 and S344) regulate proper TRF1 telomere binding. Indeed, neither TRF1-T330A nor TRF1-S344A variants were able to complement the severe proliferative defects of Trf1-deficient cells, indicating that these sites are important for TRF1 function in vivo. The observation that TRF1-T248A was able to complement Trf1 deficiency to a similar degree than wild-type TRF1, even though showed a shorter half-life in turn-over experiments, might be explained by the fact that the complementation assay was performed with overexpressed proteins thus the effects on protein stability maybe masked. Further studies using knock-in mouse models warrants further clarification of the in vivo phenotypic consequences of these mutations.

Pathologic activation of the PI3K pathway is among the most frequent signaling events associated with cellular transformation, cancer and metastasis[42, 43]. The PI3K pathway regulates a wide range of target proteins to control cell proliferation, survival, apoptosis and cell growth[40]. In the last years, pharmaceutical interest has arisen to develop strategies to inhibit PI3K signaling in cancer cells. Several PI3K downstream targets have been proposed to mediate the role of PI3K in cancer[45, 46]. Our results suggest that induction of telomere DNA damage upon PI3K inhibition constitutes another mechanism underlying the anti-cancer effects of PI3K/ATK chemical inhibitors. Indeed, treatment of mice suffering from aggressive lung cancer in vivo with the ETP-47037 PI3K/TRF1 inhibitor developed by us significantly decreased telomere TRF1 foci, induced telomeric DNA damage and blocked tumor growth as compared to non-treated mice[39]. Interestingly, PI3K inhibition enhances DNA damage in breast cancers that have genetic aberrations in BRCA1 and TP53 by impairing production of nucleotides needed for DNA synthesis and repair[82, 83]. Our work demonstrates a central role of the PI3K/AKT pathway in regulation of telomere protection, thus highlighting components of this pathway as novel targets for telomere-based therapies in cancer and age-related diseases.

## Methods

**Cell culture assays**. Immortalized Mouse embryonic fibroblasts (MEFs) p110α (lox/lox) or (+/+)[75], $Trf1^{+/+}$ $p53^{-/-}$ or $Trf1^{flox/flox}$ MEFs[29], and $K$-$Ras^{\Delta/LG12Vgeo}$ $p53^{-/-}$ tumor-derived cell line CHA-9-3 cell line[39], were cultured in DMEM supplemented with 10% fetal bovine serum (FBS).

For adenoviral infection, cells were infected twice with Adenovirus CMV-CRE-GFP. Retroviral supernatants were produced in 293T cells ($5 \times 10^5$ cells per 100 mm diameter dish) transfected with the ecotropic packaging plasmid pCL-Eco and either pBabe-Cre, pBabe-GFP or eGFP-TRF1 pWzl-Hygro (a gift from T. de Lange, Addgene plasmid #19834). Two days later, retroviral supernatant (10 ml) were collected. MEF were seeded the previous day ($8 \times 10^5$ cells per 100 mm diameter dish) and received 2 ml of each corresponding retroviral supernatant. This procedure was repeated three times at 12 h intervals.

**Generation of PDXs**. Experiments were conducted following the European Union's animal care directive (86/609/EEC) and were approved by the Ethical Committee of Animal Experimentation of the Vall d'Hebron Research Institute and patient consent. Fresh primary or metastatic human breast tumors were obtained from patients at time of surgery or biopsy. Fragments of 30–60 mm³ were immediately implanted into the mammary fat pad (surgery samples) or the lower flank (metastatic samples) of female athymic NMRI nu/nu mice. Mice were continuously treated with 17 beta-Estradiol (Sigma-Aldrich) in the drinking water at 1 μM. Upon growth of the engrafted tumors, the model was perpetuated by serial transplantation. In each passage, flash frozen and formalin-fixed paraffin embedded samples were taken for genotyping and histological studies. To evaluate the sensitivity to the drugs one tumor of each PDX model was expanded to 24 recipient mice. When the tumors reached an average volume of 200 mm³, mice were separated into two treatment arms: vehicle (0.5% methylcellulose in PBS) or BYL719 (35 mg/kg QD, 6 days/week). These doses are comparable to 75% of the human maximum tolerated dose. The treatment was prolonged for 12 days.

In all experiments, mouse weight and tumor dimensions were recorded twice weekly with a digital caliper starting with the first day of treatment. The tumor volume was calculated as $V = 4\pi/3(L \times l^2)$, "$L$" being the largest diameter and "$l$" the smallest.

**Immunofluorescence staining techniques**. TRF1 immunofluorescence of TRF1 was performed in lung tumor-derived cell line (CHA9.3) and in MEFs[39]. A rat monoclonal antibody against mouse TRF1 developed by the Monoclonal antibody unit at CNIO was used to detect endogenous TRF1 (1:500). GFP-TRF1 fluorescence was directly measured in cells fixed in paraformaldehyde. For quantification of TRF1 foci intensity, pictures of fixed cells were automatically acquired from each well by the Opera HCS system (Perkin Elmer). Sixty images of random fields per well, with a 40× magnification lens, were taken under non-saturating conditions. At least $1 \times 10^3$ cells were analyzed for each well. Briefly, images were segmented using the DAPI staining to generate masks matching cell nuclei from which-TRF1 foci were analyzed.

Formalin-fixed paraffin-embedded PDXs sections were permeabilized with 0.5% PBS-Triton for 1 h and blocked first with fetal bovine serum for 2 h and then in 5% BSA for 2 h. Samples were incubated O/N at 4 °C with the primary antibody. Antibodies used were anti-TRF1 (1:200) (TRF-78, ABCAM, ab10579), anti-RAP1 (1:300) (Bethyl Laboratories, A300-306A), and anti-phospho-histone2 H2AX-Ser139 (γH2AX) (1:300) (Millipore, 05-636). Slides were further incubated with 488-Alexa or 555-Alexa labeled secondary antibodies. Nuclei were counterstained in a 4 μg/ml 4′,6-diamino-2-phenylindole (DAPI)/PBS solution before mounting with Vectashield.

An immune-Fish was performed in formalin-fixed paraffin-embedded PDXs sections to identify telomeric TRF1 and quantify telomere length. TRF1 foci were detected by immunofluorescence as described above. Quantitative FISH was performed as follows: samples were permeabilized for 2 h in PBS-5% Triton, blocked for 1 h with 5% BSA in PBS, and immunofluorescence with TRF1 homemade rat TRF1 antibody diluted 1:500 was performed. After immunofluorescence, samples were fixed for 20 min in 4% paraformaldehide in PBS and followed by Q-FISH. Briefly, samples were washed with PBS and dehydrate in EtOH 70, 90 and 100%. The samples were then incubated with a telomeric PNA probe labeled with CY3 (Panagene) in 50% formamide for 30 min, washed in the presence of 50% formamide and counterstained with DAPI. Telomeric TRF1 foci were identified by colocalization of CY3 and 488-Alexa double positive spots. Confocal microscopy was performed at room temperature with a laser-scanning microscope (TCS SP5; Leica) using a Plan Apo 63Å-1.40 NA oil immersion objective (HCX; Leica). Maximal projection of Z-stack images generated using advanced fluorescence software (LAS) was analyzed with the Definiens XD software package. The DAPI images were used to detect signals inside the nuclei.

**Western blot analysis**. Twenty micrograms of nuclear extracts of untreated control (DMSO) or treated with inhibitors in each cased indicated cells were resolved in 4–12% SDS/PAGE gels (NuPAGE Invitrogen) and transferred to nitrocellulose membranes. Blots were incubated with the primary antibodies anti-TRF1 1:500 (home-made monoclonal Rat, CNIO; TRF-78, ABCAM, ab10579) and with anti-SMC1 1:2000 (Bethyl Laboratories #A300-055A) as loading control. For nuclear protein preparation, the Nuclear/Cytosol Fractionation Kit (K266-100, BioVision) was used according to the manufactures protocol.

For detection of AKT, PRAS40 (T246) and pS6 in CHA9.3 cells, untreated control (DMSO) or treated with the indicated inhibitors, were lysed in RIPA buffer supplemented with 100 mM pefablock (Roche Diagnostics) and protease inhibitor cocktail (Roche Molecular Biochemicals). Proteins were resolved on 10% SDS-PAGE, and transferred to nitrocellulose membrane (Amersham™ Protan™ 0.2 μm nNC). Membranes were incubated with antibodies against phospho-AKT (Ser473), AKT, phospo-S6 (Ser 235/236), and S6Ribosomal from Cell Signaling, phospo-PRAS40 (Thr246) from BIOSURCE and anti α-tubulin form Sigma, overnight at 4 °C. After washing, blots were incubated with secondary antibody: Alexa Fluor 680 goat anti-rabbit IgG (Invitrogen) or anti-mouse IgG DyLight 800 conjugated (Thermo Scientific), and visualized using an Odyssey infrared imaging system (Li-Cor Biosciences).

Sample preparation for dimerization analysis was performed in the absence of reducing agents (without dithiothreitol, $DTnT$, and 2-mercaptoetanol). Proteins were not pre-heated before gel loading.

Cycloheximide chase experiments were performed in MEFs expressing WT or mutant (T330A, S344A, T248A, or T330A/S344A) GFP-TRF1. Cells were seeded in 10 cm plates. After 16 h, cells were pre-treated with 100 μg/ml cycloheximide (Sigma) for 30 min. Cells were then collected at 0, 1, 2, 5, 4, and 5 h. In addition,

cells were treated with both cycloheximide and the proteasome inhibitor, Bortezomib (5 µM) for 5 h. Western blot was performed against TRF1 and SMC1 as decribed above.

The uncropped blot scans of this work are shown in Supplementary Figs 4 and 5.

**Electrophoretic mobility shift assay.** In vitro gel-shift assays were performed with 1 µg GST-TRF1 wt or mutant (T330A, S344A; T248A) and $^{32}$P-labeled ds (TTAGGG)7 telomeric probe. Briefly, DNA probes were prepared by annealing the two oligonucleotides, sense (5′-GGGTTAGGGTTAGGGTTAGGGTTAGGGT-TAGGGTTAGGGTTAGGGCCCCTC-3′) and antisense (5′-GAGGGGCCC-TAACCCTAACCCTAACCCTAACCCTAACCCTAACCC-3′), end-labeled with [γ-$^{32}$P]ATP (Amersham Biosciences) and T4-polynucleotide kinase (New England BioLabs) and purified by free nucleotide removal spin column (Qiagen). Labeled-DNA probes were incubated with 1 µg of purified TRF1 protein for 20 min at room temperature in a 30-µl reaction containing 20 mm HEPES-KOH, pH 7.9, 150 mm KCl, 5% (v/v) glycerol, 4% (w/v) Ficoll, 1 mm EDTA, 0.1 mm MgCl$_2$, 0.5 mm dithiothreitol, and 20 µg of bovine serum albumin, and poly (dI-dC) (1 µg) as the nonspecific competitor (Amersham Biosciences). For supershift analysis, recombinant TRF1 was preincubated with anti-TRF1 antibody (rat homemade) before EMSA. The DNA-protein complexes were resolved on a 5% non-denaturing polyacrylamide (29:1 acrylamide:bisacrylamide) gel with 0.5× Tris-borate-EDTA, TBE (Sigma) as running buffer. Gels were dried under a vacuum at 80 °C and auto radiographed.

**RNA and qPCR.** Total RNA *Trf1* expression was quantified by qPCR using the following primers: TRF1 forward (5′-TCTAAGGATAGGCCAGATGCCA-3′) and TRF1 reverse (5′-CTGAAATCTGATGGAGCACGTC-3′). GAPDH was used as the housekeeping gene. We determined the relative expression of *Trf1* in each sample by calculating the 2ΔCT value. For each sample, 2ΔCT was normalized to the control 2ΔCT mean.

**Q-FISH on metaphase spreads.** Lung cancer-derived cell line (CHA-9-3) and immortalized *P110α*$^{lox/lox}$ MEFS were treated either with 10 µM of ETP-47037, ETP-47228, or AKTi (MK-2206) for 24 h. Q-FISH in metaphase spreads was performed as described above, but increasing the concentration of formamide in both the incubation and the washes to 70%.

**In vitro TRF1 phosphorylation.** Full-length mouse TRF1 was cloned using the Gateway™ technology (Thermo Fisher Scientific) into the expression vector pDEST565, which adds two tags (6xHis and GST) at the N terminus of the encoded protein. Protein was expressed in *Escherichia coli* strain BL21(DE3) and purified by affinity chromatography using a Ni$^{2+}$ column (HisTrap FF crude, 17-5286-01, GE Healthcare) in an AKTA Prime system (GE Healthcare), followed by dialysis against 20 mM phosphate buffer pH 7.5, 200 mM NaCl, 2 mM TCEP.

One µg of purified GST-TRF1 was incubated with 0.2 µg of human GST-AKT1 (1379-0000-2, ProQinase) with or without AKT inhibitor (MK-2206, 25 µM) in kinase buffer (50 mM Hepes pH 7.9, 100 µM ATP, 10 mM MgCl$_2$, 5 mM DTT) containing 5 µCi [γ-$^{32}$-P]ATP in a total volume of 25 µl. The peptide RBER-GSK3 (14-27) (0349-0000-5, ProQinase) (100 ng) was used as positive control. The reactions were performed at 30 °C for 1 and stopped by addition of Lamely buffer (6×). Samples were resolved in 4–12% SDS-PAGE gels and subjected to autoradiography.

**Generation of *GST-TRF1* and *eGFP-TRF1* mutant alleles.** For site-directed mutagenesis, QuickChange lightning multi site-directed mutagenesis (210513, Agilent Technologies) was used. In brief, PCRs were performed following the manufacturer's protocol with either the pDEST565-mTRF1 expression vector or the eGFP-TRF1 pWzl-Hygro retroviral vector as templates and the following PAGE purified mutagenic primers (Sigma-Aldrich):

mTRF1-T330A: 5′-GAACGAAGCAAGAACAGGAGCTCTTCAGTGTGAAA CAAC-3′.

mTRF1-S344A: 5′-GGAAAGGAACCGAAGAACCGCTGGAAGGAATAGAT TGTGT-3′.

mTRF1-T248A: 5′-CAACTTTTCTAATGAAGGCAGCAGCAAAAGTAGTG GAAAATGAGAAA-3′.

PCR products were digested with *Dpn I* restriction enzyme to digest the parental (non-mutated) DNA for 1 h at 37 °C and then, transformed into XL-10-Gold® ultracompetent cells. Individual colonies were grown and DNA extracted with QIAprep Spin Miniprep Kit (27106, QIAGEN). Mutations were confirmed by sequencing with a specific TRF1 primer 5′-TTCCACTCCCTTTTCCAACACT-3′. Finally, 50 ng of each mutant DNA was used to transform BL21(DE3). Protein production and phosphorylation of the respective mutant GST-TRF1 protein were carried out following the same protocols described above.

**Identification of TRF1 phosphopeptides by LC/MS/MS analysis.** One µg of purified GST-TRF1 was incubated with 0.2 µg of human GST-AKT1 (1379-0000-2, ProQinase) in kinase buffer (50 mM Hepes pH 7.9,100 µM ATP, 10 mM MgCl$_2$, 5

mM DTT) in a total volume of 25 µl for 1 h at 30 °C. Protein samples were diluted with 9 M urea in 50 mM ammonium bicarbonate (ABC) and subsequently reduced and carbamidomethylated in 15 mM tris(2-carboxyethyl)phosphine (TCEP) and 30 mM chloroacetamide for 45 min at 25 °C protected from light. After diluting the urea to 2 M with 50 mM ABC, samples were digested overnight with Lys-C (1:50 enzyme/protein w/w), diluted to 1 M urea, and further digested with Trypsin (1:50 enzyme/protein w/w) for 6 h at 37 °C. Resulting peptides were desalted by home-made C18 Empore tips and analyzed by LC-MS/MS onto a LTQ-OrbitrapVelos instrument. The raw files were processed using the Proteome Discoverer 1.4.0.1 software. Fragmentation spectra were searched against the mouse Uni-prot_KB/Trembl Database (43539 entries), supplemented with a home-made database comprising the contaminant proteins most commonly found in our assays, using Sequest as the search engine. The precursor and fragment mass tolerances were set to 20 p.p.m. and 0.5 Da, respectively, and up to two tryptic missed cleavages were allowed. Carbamidomethylation of cysteine was considered as fixed modification, while oxidation of methionine, and phosphorylation of serine, threonine, and tyrosine were chosen as variable modification for database searching. The results were filtered to 1% false discovery rate (FDR) using percolator.

**Synthetic protocols for ETP compounds.** The HPLC-MS measurements were performed using a HP 1100 from Agilent Technologies. Method LCM1: Gemini-NX C18 (100 × 2.9 mm; 5 mm). Solvent A: water with 0.1% formic acid. Solvent B: acetonitrile with 0.1% formic acid. Gradient: 5% of B to 100% of B within 8 min at 50 °C, DAD. The MS detector was configured with an electrospray ionization source or API/APCI. Nitrogen was used as the nebulizer gas. [M + 1]$^+$ means the protonated mass of the compound.

NMR spectra were recorded in a Bruker Advance II 300 spectrometer.

ETP-47228: the compound was synthesized following similar synthetic procedure reported in WO2010119264 for compound 2-243.

LCMS1: retention time: 2.98 min, [M + 1]$^+$ = 555.1.

$^1$H NMR (300 MHz, DMSO) δ 8.91 (s, 1H), 8.88 (s, 1H), 8.18 (s, 1H), 8.02 (d, *J* = 8.7 Hz, 2H), 7.54 (d, *J* = 8.7 Hz, 4H), 7.52 (d, *J* = 8.7 Hz, 2H), 7.35 (s, 1H), 7.33 (d, 2H), 4.27 (m, 4H), 3.78 (m, 4H), 3.49 (wide s, 4H), 2.50 (s, 3H), 2.31 (m, 4H), 2.19 (s, 3H).

ETP-50952: the compound was synthesized following the similar synthetic procedure used for synthesis of ETP-47228.

LCMS1: retention time: 3.26 min, [M + 1]$^+$ = 553.1.

$^1$H NMR (300 MHz, DMSO) δ 8.93 (s, 1H), 8.88 (s, 1H), 8.10 (s, 1H), 8.01 (d, *J* = 8.7 Hz, 2H), 7.55 (d, *J* = 8.7 Hz, 2H), 7.52 (d, *J* = 8.7 Hz, 2H), 7.34 (d, *J* = 8.4 Hz, 2H), 7.33 (s, 1H), 4.28 (m, 4H), 3.49 (m, 4H), 2.50 (s, 3H under DMSO-d6 signal), 2.31 (m, 4H), 2.20 (s, 3H), 1.67 (m, 6H).

ETP-47037: synthetic procedure and analytical data are reported in WO2011089400 (compound 3–10 of the patent).

ETP-51259: the compound was synthesized following the similar synthetic procedure used for synthesis of ETP-47037.

LCMS1: retention time: 1.94 min, [M + 1]$^+$ = 472.6.

$^1$H NMR (300 MHz, DMSO) δ 8.85 (s, 2H), 8.58 (s, 1H), 6.82 (s, 3H), 3.77 (m, 6H), 3.12 (m, 4H), 2.87 (s, 3H), 2. 55 (m, 4H), 1.67 (m, 6H).

**Synthesis of ETP-46992.** The synthesis of ETP-46992 is described in ref. [66].

**Commercial inhibitors used in the study.** The commercially available PI3K/mTOR and AKT inhibitors, GDC-0941, BKM-120, BYL-719, TGX-221, BEZ-235, GSK-2126458, and MK-2206, described in the manuscript were purchased from Selleckchem (www.selleckchem.com). The inhibitors were tested at 10 µM except GSK-2126458 that was used at 1.0 µM.

**PI3K alpha biochemical assay.** The kinase activity was measured by using the commercial ADP Hunter™ Plus assay available from DiscoveRx (#33-016), which is a homogeneous assay to measure the accumulation of ADP, a universal product of kinase activity. The enzyme, PI3K (p110α) was purchased from Cama Bios-ciences (#07CBS-0402A). The assay was performed following the manufacturer recommendations. To calculate the IC$_{50}$ of the selected compounds, serial 1:5 dilutions were used in a concentration range from 128 pM to 50 µM fluorescence counts were read in a Victor instrument (Perkin Elmer) with the recommended settings (544 and 580 nm as excitation and emission wavelengths, respectively). Values were normalized against the control activity included for the enzyme (Le., 100 % PI3 kinase activity, without compound). These values were plotted against the inhibitor concentration and were fit to a sigmoid dose-response curve by using Activity base software from IDBS.

**PI3K isoforms biochemical assay.** PI3Kα (p110α/p85α), PI3K-δ (p110δ/p85α), and PI3K-γ (p110γ) were purchased from Invitrogen (PV4789, PV 5274, and PV4787, respectively). PI3K-β (p110β/p85α) was obtained from Millipore (#14-603). The kinase activity was measured with the commercial PI3-kinase (human) HTRF™ assay available from Millipore (#33-017), following the manufacturer recommendations. The final enzyme concentration in the assay were 100 pM for PI3K-α and PI3K-δ, 500 pM for PI3K-β, and 4 nM for PI3K-γ. To calculate the

IC$_{50}$ of the selected compounds, serial 1:5 dilutions were used in a concentration range from 12 pM to 25 μM. The enzyme was preincubated with the inhibitor and with 10 μM of PIP2 for 5 min. The enzymatic reaction was started by addition of ATP at 50 times the Km. The incubation was kept at 37 °C for 60 min. Stop solution and detection mix were added sequentially to the wells and plates were incubated 3 h in the dark at 25 °C. HTRF counts were measured in an EnVision reader (Perkin Elmer) using the recommended settings. Values were normalized against the control activity included for the enzyme (100% PI3 kinase activity, without compound). These values were plotted against the inhibitor concentration and were fit to a sigmoid dose-response curve by using Activity base software from IDBS.

**mTOR biochemical assay**. mTOR biochemical assay was performed with a LanthaScreen™ kinase activity assay form invitrogen. The kinase mTOR (FRAP1; ThermoFisher, PV4753), a GFP-labeled substrate (GFP-4E BP1, ThermoFisher, PV4759), and ATP were allowed to react. Then EDTA was added to stop the reaction and terbium-labeled antibody, Tb-anti-p4EBP1 (phosphorthreonine 46) (ThermoFisher, PV4758) to detect phosphorylated product. in a LanthaScreen™ kinase reaction, the antibody associates with the phosphorylated GFP-labeled substrate resulting in an increased TR-FRET value. The TR-FRET value is a dimensionless number that is calculated as the ratio of the acceptor (GFP) signal to the donor (terbium) signal. The amount of antibody that is bound to the tracer is directly proportional to the amount of phosphorylated substrate, so that kinase activity can be measured by an increase in the TR-FRET value. To calculate the IC$_{50}$ of the selected compounds, serial 1:3 dilutions were used in a concentration range from 1.49 nM to 10 μM. Fluorescence counts were read in an Victor reader (Perkin Elmer) with the recommended settings (495 and 520 nm as emission wavelengths for Terbium and GFP, respectively). Values were normalized against the control activity included for the enzyme (100% mTOR activity, without compound). These values were plotted against the inhibitor concentration and were fit to a sigmoid dose-response curve by using Activity base software from IDBS.

**Data availability**. The authors declare that the data supporting the findings of this study are available from the authors.

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

## Acknowledgements

We are indebted to D. Megias for microscopy analysis, to D. Calvo for protein purification as well as to J. Muñoz and F. García for LC/MS/MS analysis. The research was funded by project SAF2013-45111-R of Societal Changes Program of the Spanish Ministry of Economics and Competitiveness (MINECO) co-financed through the European Fund of Regional Development (FEDER), Fundación Botín, Banco Santander (Santander Universities Global Division) and Worldwide Cancer Research (WCR 16-1177).

## Author contributions

M.A.B., P.M. and M.M.-P. performed experiments and wrote the manuscript. C.B., E.G.-C., A.B.G. and J.P. performed the chemical studies of the PI3K/AKT inhibitors. J.M.-T. performed TRF1 protein purification. M.P., J.C. and V.S. provided PDX samples.

## Additional information

**Competing interests:** The authors declare no competing financial interests.

