## [Peer Review File · Nature Communications]

Reviewers' Comments:

Reviewer #1:

Remarks to the Author:

The work of Méndez-Pertuzet al. entitled "Modulation of telomere protection by the PI3K/AKT pathway" reports the regulation of the Telomeric Repeat Factor 1 (TRF1), which controls telomere length as a component of the shelterin nucleoprotein complex by the PI3K/AKT signaling pathway via direct phosphorylation by AKT. The authors use an elegant chemical as well as a genetic approach to show the implication of PI3K α in the regulation of TRF1 protein level, they identify TRF1 as a direct substrate of AKT and demonstrate the in vivo relevance by using a breast cancer patient-derived PDX model. This is a very timely and mostly well written article with intriguing implications which might be of great interest to the readers of Nature Communications. However, important gaps in the understanding of the underlying molecular mechanisms and questions concerning their clinical relevance remain to be addressed. Therefore, the manuscript would greatly benefit from the development of several aspects.

Major Points:

1. The presented data suggests that inhibition/ablation of PI3K α and subsequent decrease of AKT-mediated TRF-1 phosphorylation results in a decrease of the total amount of TRF-1 protein but the link between post-translational modification and protein level remains to be established. The authors should perform experiments to shed light on the mechanism investigating the effect of PI3K/AKT pathway inhibition and TRF-1 phosphorylation on TRF-1 synthesis and degradation using cycloheximide chase and/or fluorescence recovery after partial photobleaching experiments with TRF-1 phosphomimetic and non-phosphorylatable mutants in the presence or absence of inhibitors of the proteasome.
2. In addition, the authors should determine if TRF-1 phosphorylation affects TRF-1 dimerization and DNA binding
3. The data presented in the current manuscript would predict tumour samples with high level of AKT activation (using AKT-Ser473 as a biomarker) to correlate with higher level of TRF-1 protein compared with samples from the same tumour type with lower AKT-Ser473 phosphorylation. Importantly, this should affect telomere length. In order to validate the clinical relevance of the findings, the authors should try to validate their data in a significant number of patient samples.

Minor points:

1. The authors use micromolar concentrations of potent kinase inhibitors without providing their specificity profile. Therefore, it can't be ruled out that their effect is due to the inhibition of other ATP-dependent enzymes.
2. Compound concentrations should be provided for all experiments and included in the figure legends
3. As an in-house antibody against TRF-1 has been used, data on its validation should be provided
4. Figure legends are often incomplete. E. g. it is not clear to what TRF-1 inhibition refers to in Figure 1D. What is the difference between the three western blot panels in Figure 6A, ect
5. The data on GSK3a/b-mediated rescue are interesting but no mechanism has been proposed to explain it. In order to focus on a clear message, these data should be either removed or further developed (several GSK3a/b substrates are known and their effect on TRF-1 could be explored)
6. It is not clear how the quantification of multitelomeric signals (MTS) has been performed. This information should be provided
7. It is unclear why the remaining signal for AKT-mediated phosphorylation of the TRF-1 with triple substitution T248A/T330A/S344A is stronger than the signal from TRF-1 with single substitutions in Figure 7G.
8. The fact that there is a detectable signal in the absence of the identified AKT phosphorylation site suggests the presence of additional sites within TRF-1 which could contribute to the observed phenotype
9. As TRF1 spot intensity by immunofluorescence is a very indirect way to measure TRF1 protein level (e.g. spot intensity can decrease by changes in subcellular TRF1 localization), the authors should perform western blot analysis of BYL719/vehicle treated PDXs
10. The discussion is rather an extended abstract than a critical discussion of the results and

should be revised

11. There are some typos such as Pi3K instead of PI3K ect

Reviewer #2:

Remarks to the Author:

The manuscript by Mendez-Pertuz et al. reports on the effect of PI3K/AKT pathway inhibition on telomeric factor TRF1 localisation. The authors show that treatment of mouse cells with a panel of PI3K/AKT inhibitors triggers reduction of TRF1 foci intensity and protein expression. GSK3 inhibitors reverse the foci intensity, but not TRF1 protein downregulation. Using genetic approaches, reduction in TRF1 signal is attributed to the p110 α catalytic subunit of PI3K. Direct phosphorylation of TRF1 by AKT is shown in vitro and in cell extracts using mass spectrometry and two residues are mapped. Finally, AKT inhibitors cause a decrease in TRF1 signal in patient-derived xenografts.

The novelty of this study is partial, as AKT-dependent phosphorylation of TRF1 was previously shown (Chen, Teng & Wu, Cancer Invest. 2009). Whilst the previously published data indicated that this phosphorylation results in telomere shortening, the current manuscript suggests that it protects against telomere damage and fragility. This inconsistency should be addressed, at least in Discussion.

The fundamental problem with the current study is that out of the two locally synthesized compounds (ETP-47037 and ETP-47228) only one, ETP-47037, effectively abolished Ser437 AKT phosphorylation (Fig. 1; see below). Both inhibitors were reported in a previous study from the same lab (Garcia-Beccaria et al., EMBO Mol Medicine 2015) to inhibit binding of TRF1 to telomeres. Consistent with this, in the current study both compounds decreased TRF1 immunofluorescence signal and protein stability, however only one of them effectively inhibited PI3K/AKT pathway (Ser473 AKT phosphorylation as a functional assay). Therefore, the claim that TRF1 reduction is attributable to AKT pathway inhibition is not substantiated.

The data generated with GSK3 inhibitors are impossible to interpret: GSK3i rescues the TRF1 signal reduction induced by AKTi, but not the reduction in protein levels. No mechanistic explanation is provided for this apparent paradox. Further adding to the confusion is the claim that AKT phosphorylates TRF1 and GSK3, but GSK3 does not phosphorylate TRF1. Yet, inhibiting GSK3 can reverse the effect of AKT on TRF1. How could these results be integrated into a coherent model for AKT-dependent regulation of telomeric function remains unclear. The role of GSK3 in WNT signalling is not mentioned in the manuscript.

Specific comments:

Fig. 1C:

- The data for ETP-47228 inhibition of Ser473 AKT phosphorylation (and the negative effect of ETP-50952 counterpart) are weak relative to ETP-47037. At 24h and 10 μ M ETP-47228 did not inhibit at all Ser473 AKT phosphorylation, which is also in contrast with the panel of commercial inhibitors shown in Fig. 2C. It is unlikely that the results obtained with the ETP-47228 can be attributed to PI3K/AKT inhibition.
- In order to convincingly show inhibition of PI3K/AKT pathway, phosphorylation of other downstream targets should be examined.

In Fig. 1D:

- Does 'PI3K activity' under the graph refer inhibition of PI3K activity or its inhibition? Same for the graph in Fig. 2D.

Fig. 7A:

- MW markers should be included here because it is unlikely that the bands shown correspond to full length TRF1 (approx. 66 KDa) and GSK3 peptide 13 amino acid long (less than 2 KDa).
- Phosphorylation of the GSK3 peptide appears independent of AKT, because AKTi does not abrogate it. What is the authors explanation of this result?
- Whether GSK3 can phosphorylate TRF1 should also be addressed in this experiment, in order to demonstrate the specificity of the effect to AKT.

Fig. 7B,C:

- This figure is confusing; the graphs should be labelled with the conditions of which experiment is done. Figure legend is incorrect.

Fig. 8:

- The authors are examining here a dominant negative phenotype, since endogenous TRF1 is expressed in the transduced cells (p53^{-/-} MEFs and CHA 9-3 mouse cancer cells). A meaningful experiment would be to transduce these constructs into MEFs carrying a conditional Trf1 deletion. This would eliminate any possible interference of endogenous TRF1.

Fig. 9:

- In this figure the authors use BYL-719, a commercial PI3Ka inhibitor and not one of the 'proprietary' compounds used in the other figures; no explanation is provided.
- It is not clear that the 53BP1 and gH2AX quantification refers to telomere damage. The compound used here could introduce non-specific DNA damage anywhere in the genome.
- The co-localisation of RAP1 and gH2AX signal in 9G is minimal.

Point-by-point answer to the reviewers NCOMMS-17-08869-T

[AUTHORS] We would like to begin our point-by-point answer to the reviewer's comments by thanking you and the reviewers for considering our work "***of considerable potential interest***". We are glad that the reviewers appreciate our data and manuscript. We also appreciate the reviewer's suggestions and their interest in our work, and strongly believe that many of the raised points are useful and will help us to improve our original manuscript.

We would like to take the opportunity to highlight the following statement by reviewer #1:

"The authors use an elegant chemical as well as a genetic approach to show the implication of PI3K α in the regulation of TRF1 protein level, they identify TRF1 as a direct substrate of AKT and demonstrate the in vivo relevance by using a breast cancer patient-derived PDX model. This is a very timely and mostly well written article with intriguing implications."

ANSWER to Reviewers' Comments:

Detailed Answer to Reviewer #1

[REVIEWER] The work of Méndez-Pertuzet al. entitled "Modulation of telomere protection by the PI3K/AKT pathway" reports the regulation of the Telomeric Repeat Factor 1 (TRF1), which controls telomere length as a component of the shelterin nucleoprotein complex by the PI3K/AKT signaling pathway via direct phosphorylation by AKT. The authors use an elegant chemical as well as a genetic approach to show the implication of PI3K α in the regulation of TRF1 protein level, they identify TRF1 as a direct substrate of AKT and demonstrate the in vivo relevance by using a breast cancer patient-derived PDX model. This is a very timely and mostly well written article with intriguing implications which might be of great interest to the readers of Nature Communications. However, important gaps in the understanding of the underlying molecular mechanisms and questions concerning their clinical relevance remain to be addressed. Therefore, the manuscript would greatly benefit from the development of several aspects.

[AUTHORS] We would like to thank this reviewer for the thorough revision of our manuscript and for considering that we "*use an elegant chemical as well as a genetic approach to show the implication of PI3K α in the regulation of TRF1 protein level, they identify TRF1 as a direct substrate of AKT and demonstrate the in vivo relevance by using a breast cancer patient-derived PDX model. This is a very timely and mostly well written article with intriguing implications which might be of great interest to the readers of Nature Communications*". The reviewer also has a number of insightful suggestions which we have addressed in full and that we think have greatly contributed to improve this manuscript.

[REVIEWER] The presented data suggests that inhibition/ablation of PI3K α and subsequent decrease of AKT-mediated TRF-1 phosphorylation results in a decrease of the total amount of TRF-1 protein but the link between post-translational modification

and protein level remains to be established. “The authors should perform experiments to shed light on the mechanism investigating the effect of PI3K/AKT pathway inhibition and TRF-1 phosphorylation on TRF-1 synthesis and degradation using cycloheximide chase and/or fluorescence recovery after partial photobleaching experiments with TRF-1 phosphomimetic and non-phosphorylatable mutants in the presence or absence of inhibitors of the proteasome.”

[AUTHORS] As suggested by the reviewer, we have performed cycloheximide chase experiments in MEFs to determine the stability and degradation of both wild-type eGFP-TRF1 (WT) and the non-phosphorylatable eGFP-TRF1 mutants in the presence or absence of proteasome inhibitors. The results show that the mutant proteins present a two-fold reduction in their half-life (HL<2.5 hours) as compared to wild-type TRF1 (HL=4.5 hours), indicating that AKT-mediated phosphorylation is involved in TRF1 stability (**new Fig. 7A**) (see page 2, Abstract; page 17, 3rd paragraph; page 18, 1nd paragraph; page 23, 1st paragraph).

[REVIEWER] The authors should determine if TRF-1 phosphorylation affects TRF-1 dimerization and DNA binding.

[AUTHORS] As suggested by the reviewer, to address the dimerization ability of the TRF1 mutants versus wild-type TRF1, we performed Western blot analysis in non-reducing conditions to analyze the dimerization pattern of TRF1 mutants compared to wild-type TRF1. The results clearly show that none of the non-phosphorylatable mutations under study affect the dimerization capacity of TRF1 (**new Fig. 7B**)(page 18, 2nd paragraph).

As suggested by the reviewer, we have also performed Electrophoretic mobility shift assay (EMSA) to determine the DNA-binding ability of the different TRF1 mutants vs wild-type TRF1. The results show that both wild-type TRF1 and the TRF1^{T248A} mutant efficiently bind telomeric DNA repeats, while both the TRF1^{T330A} and TRF1^{S344A} mutants bind telomeric DNA repeats less efficiently, thus indicating that these AKT-phosphorylation sites are important for TRF1 binding to telomeric repeats (**new Fig. 7C**)(see page 2, Abstract; page 18, 3rd paragraph; page 23, 1st paragraph).

[REVIEWER] The data presented in the current manuscript would predict tumour samples with high level of AKT activation (using AKT-Ser473 as a biomarker) to correlate with higher level of TRF-1 protein compared with samples from the same tumour type with lower AKT-Ser473 phosphorylation. Importantly, this should affect telomere length. In order to validate the clinical relevance of the findings, the authors should try to validate their data in a significant number of patient samples.

[AUTHORS] We thank the reviewer for this suggestion since we think the new data considerably improves our work. In the revised manuscript, we have independently studied each PDX (7 independent PDXs untreated or treated with BYL719 in duplicate) of the same tumor type (breast tumors). In particular, in each PDX we have analyzed: p-AKT, TRF1 expression, and telomere length. Importantly, to specifically quantify TRF1 present at telomeres, we have performed a TRF1 immunofluorescence followed by a telomeric DNA quantitative q-FISH to co-localize TRF1 to telomeres. We then quantitatively analyzed the intensity of telomeric TRF1 foci that colocalized with telomeric DNA. We have also used the quantitative telomere qFISH to determine telomere length in each PDX. In addition, we quantitatively analyzed p-AKT (S473) levels in serial sectioned paraffin-embedded PDXs. In the revised manuscript we now show all these new results for independent PDXs (**new Fig. 8A-F**). We observe a

significant correlation between p-AKT (S473) and TRF1 levels (**new Fig. 8F**). Importantly, untreated PDXs with the lowest basal p-AKT levels (undetectable) present the lowest telomeric TRF1 levels (see data in **new Fig. 8B,D** regarding PDX131 and PDX39). Of note, the PDXs showing the highest p-AKT level is also the one presenting the highest telomeric TRF1 level (see data in **new Fig. 8B,D** regarding BYL719 treated PDX131). Analysis of telomere length by q-FISH in these PDX samples also revealed a correlation between TRF1 levels and mean telomere length, although the correlation was lower than with p-AKT (**new Fig. 8F**). We now discuss all these new data in the revised manuscript text (**page 19-20; page 21, 2nd paragraph; 22, 1st paragraph**).

Minor points:

[REVIEWER] The authors use micromolar concentrations of potent kinase inhibitors without providing their specificity profile. Therefore, it can't be ruled out that their effect is due to the inhibition of other ATP-dependent enzymes.

[AUTHORS] To answer to this commentary by the reviewer, we have now included in the revised manuscript a kinase selectivity profile information for ETP-47037 and ETP-47228 at 1 μ M (**new Supplementary Table 1**). We have also included in the revised manuscript a paragraph referencing published data on the inhibitors used in this study (**page 9, 2nd paragraph**). In particular, careful analysis of both published data and CNIO's own data on the kinase selectivity of the PI3K inhibitors used in the present study reveals a high selectivity of these compounds for the inhibition of PI3K isoforms and/or mTOR (see Fig. 2A, B). Thus, we can rule out potential off-targets effects contributing to the observed TRF1 modulation.

Furthermore, we think that the selected PI3K inhibitors, showing a structural diversity and differential PI3K isoforms/mTOR profiles, justify the conclusion that the main driver of the observed TRF1 modulation is the inhibition of the main target(s) for such a type of inhibitors (PI3K). In particular, this kind of chemical validation of a given phenotype using a set of structural diverse compounds sharing a common target is widely accepted as a proof that the common target is likely to be responsible for the observed effect (in the current case PI3K and TRF1 modulation).

Importantly, the genetic validation where elimination of PI3K-alpha is clearly linked to the observed TRF1 phenotype fully supports in addition the main conclusion of the present work: PI3K inhibition modulates TRF1.

[REVIEWER] Compound concentrations should be provided for all experiments and included in the figure legends

[AUTHORS] We have included throughout the revised manuscript the compound concentrations used for each particular case including the figure legends.

[REVIEWER] As an in-house antibody against TRF-1 has been used, data on its validation should be provided.

[AUTHORS]. The Monoclonal Antibody Unit at CNIO has performed an extensive validation analysis of this Rat anti-TRF1 (see **TRF1 validation file attached for reviewers**). Furthermore, we have extensively used this antibody in previous publications of the laboratory (Marión et al., Stem Cell Reports, 2017; Povedano et al., Cell Reports 2015, Garcia-Beccaria et al., 2015).

[REVIEWER] Figure legends are often incomplete. E. g. it is not clear to what TRF-1 inhibition refers to in Figure 1D. What is the difference between the three western blot panels in Figure 6A, ect

[AUTHORS] We appreciate the reviewer's comment, and we will clarify the figure legends throughout the revised manuscript. In particular, Fig. 1D shows inhibition of TRF1 foci by immunofluorescence in lung cancer cells treated with either ETP-47037 and ETP-47228 active compounds or their corresponding inactive analogues (ETP-51259 and ETP-50952, respectively) normalized to TRF1 inhibition levels upon DMSO treatment. This has been rephrased in the revised manuscript.

Regarding previous Fig. 6A: the three Western blots correspond to three independent experiments treated in the same conditions. For clarity purposes, we have simplified this panel and only include one representative image of other Western blots in the revised manuscript. As requested by this reviewer, the information provide in the previous version of the MS regarding GSK3 has been removed (see below). Therefore, we included new representative TRF1 WB images (**revised Fig. 4**).

[REVIEWER] The data on GSK3a/b-mediated rescue are interesting but no mechanism has been proposed to explain it. In order to focus on a clear message, these data should be either removed or further developed (several GSK3a/b substrates are known and their effect on TRF-1 could be explored)

[AUTHORS] We agree with the reviewer in that the GSK3 data presented in the previous version of the manuscript needs further development. Thus, as suggested by the reviewer, we have removed it in the revised manuscript.

[REVIEWER] It is not clear how the quantification of multitelomeric signals (MTS) has been performed. This information should be provided

[AUTHORS] Multitelomeric signals (MTS) are analyzed by Q-FISH using a telomeric probe on metaphase spreads. MTS are readily visualized by presenting a multi-dot pattern at chromosome ends in contrast to normal telomeres that show a single dot at each chromosome end (see representative image in Fig. 3E). This has been explained in the revised manuscript (**page 12, 2nd paragraph**).

[REVIEWER] It is unclear why the remaining signal for AKT-mediated phosphorylation of the TRF-1 with triple substitution T248A/T330A/S344A is stronger than the signal from TRF-1 with single substitutions in Figure 7G.

[AUTHORS] We have now performed this experiment 4-5 independent times. The quantification of the signal from these experiments is represented in the **revised Fig. 5G**, showing that the remaining phosphorylated TRF1 levels are similar in the single substitution (T330A and S344A) as in the triple substitution mutant (T330A/S344A/T248A). The comparison among the mutants are not statistically significant. We have included the p values in the revised manuscript. We have now included a new representative image of the assay (see **revised Fig. 5G**).

[REVIEWER] The fact that there is a detectable signal in the absence of the identified AKT phosphorylation site suggests the presence of additional sites within TRF-1 which could contribute to the observed phenotype

[AUTHORS] We agree with the reviewer and we have discussed this possibility in the revised manuscript text (**page 15, 2nd paragraph**).

[REVIEWER] As TRF1 spot intensity by immunofluorescence is a very indirect way to measure TRF1 protein level (e.g. spot intensity can decrease by changes in subcellular TRF1 localization), the authors should perform western blot analysis of BYL719/vehicle treated PDXs

[AUTHORS] In the revised manuscript, to specifically identify and quantify TRF1 protein at telomeres, we have performed a TRF1 immunofluorescence followed by a telomeric DNA q-FISH to co-localize TRF1 with telomeric DNA. We then quantitatively analyzed the intensity of telomeric TRF1 foci that colocalized with telomeric DNA. We strongly favor immunoTRF1-telomeric FISH to quantify the amount of telomeric TRF1 over Western blot analysis when using tissue samples, as immuno-FISH can determine telomeric TRF1 levels in a per cell basis. Several published works from our lab and others demonstrate that immuno-FISH is an accurate method for telomeric-TRF1 quantification (Martinez et al., G&D, 2009; Marion et al., Stem Cell Reports, 2017; Garcia-Beccaria et al., EMBO Mol. Med, 2015; Povedano et al., Cell Reports, 2015; Schneider et al., Ncomms, 2013).

[REVIEWER] The discussion is rather an extended abstract than a critical discussion of the results and should be revised. There are some typos such as Pi3K instead of PI3K ect

[AUTHORS] We have revised the discussion as indicated above. We have corrected the typos throughout the revised manuscript.

Detailed Answer to Reviewer #2

[REVIEWER] The manuscript by Mendez-Pertuz et al. reports on the effect of PI3K/AKT pathway inhibition on telomeric factor TRF1 localisation. The authors show that treatment of mouse cells with a panel of PI3K/AKT inhibitors triggers reduction of TRF1 foci intensity and protein expression. GSK3 inhibitors reverse the foci intensity, but not TRF1 protein downregulation. Using genetic approaches, reduction in TRF1 signal is attributed to the p110 α catalytic subunit of PI3K. Direct phosphorylation of TRF1 by AKT is shown in vitro and in cell extracts using mass spectrometry and two residues are mapped. Finally, AKT inhibitors cause a decrease in TRF1 signal in patient-derived xenografts.

The novelty of this study is partial, as AKT-dependent phosphorylation of TRF1 was previously shown (Chen, Teng & Wu, *Cancer Invest.* 2009). Whilst the previously published data indicated that this phosphorylation results in telomere shortening, the current manuscript suggests that it protects against telomere damage and fragility. This inconsistency should be addressed, at least in Discussion.

[AUTHORS] We thank the reviewer for this suggestion since we think the new data considerably improves our work. In the revised manuscript, we have independently studied each PDX (7 independent PDXs untreated or treated with BYL719 in duplicate) of the same tumor type (breast tumors). In particular, in each PDX we have analyzed: p-AKT, TRF1 expression, and telomere length. Importantly, to specifically quantify TRF1 present at telomeres, we have performed a TRF1 immunofluorescence followed by a telomeric DNA quantitative q-FISH to co-localize TRF1 to telomeres. We then quantitatively analyzed the intensity of telomeric TRF1 foci that colocalized with telomeric DNA. We have also used the quantitative telomere qFISH to determine telomere length in each PDX. In addition, we quantitatively analyzed p-AKT (S473) levels in serial sectioned paraffin-embedded PDXs. In the revised manuscript we now show all these new results for independent PDXs (**new Fig. 8A-F**). We observe a significant correlation between p-AKT (S473) and TRF1 levels (**new Fig. 8F**). Importantly, untreated PDXs with the lowest basal p-AKT levels (undetectable) present the lowest telomeric TRF1 levels (see data in **new Fig. 8B,D** regarding PDX131 and PDX39). Of note, the PDXs showing the highest p-AKT level is also the one presenting the highest telomeric TRF1 level (see data in **new Fig. 8B,D** regarding BYL719 treated PDX131). Analysis of telomere length by q-FISH in these PDX samples also revealed a correlation between TRF1 levels and mean telomere length, although the correlation was lower than with p-AKT (**new Fig. 8F**). We now discuss all these new data in the revised manuscript text (**page 19-20**) We also discussed in the discussion the findings by Chen *et al* and other groups about the role of TRF1 in telomere length regulation (**page 21, 2nd paragraph; page 22, 1st paragraph**).

[REVIEWER] The fundamental problem with the current study is that out of the two locally synthesized compounds (ETP-47037 and ETP-47228) only one, ETP-47037, effectively abolished Ser437 AKT phosphorylation (Fig. 1; see below). Both inhibitors were reported in a previous study from the same lab (Garcia-Beccaria et al., *EMBO Mol Medicine* 2015) to inhibit binding of TRF1 to telomeres. Consistent with this, in the current study both compounds decreased TRF1 immunofluorescence signal and protein stability, however only one of them effectively inhibited PI3K/AKT pathway

(Ser473 AKT phosphorylation as a functional assay). Therefore, the claim that TRF1 reduction is attributable to AKT pathway inhibition is not substantiated.

[AUTHORS] As suggested by the reviewer, in the revised manuscript we have performed a more thorough time-course inhibition profile including intermediate time points between 4h and 24h of AKT phosphorylation at Ser473 by ETP-47037, ETP-51259, ETP-47228 and ETP-50952 at 10 μ M in CHA-9.3 cell line. We include these new data in the revised manuscript (**new Fig. 1C; new Supplementary Fig. 1**). The results show that the inhibitory activity of ETP-47037 was more prolonged than that of ETP-47228. Thus, ETP-47037 mediated p-AKT S473 inhibition was maintained during the 24-hour treatment while the ETP-47228 mediated inhibition decreased 35% after 24 hours as compared to 1 hour treatment (**new Fig. 1C**). ETP-47228 (10 μ M) is able to inhibit Ser473 AKT phosphorylation clearly at 4h similarly to ETP-47037. It is reasonable that ETP-47228-mediated inhibition is sufficient to exert its effect on TRF1 modulation after 24h. We also performed a time course inhibition profile of two PI3K downstream effectors, PRAS40 (T246) and P6 (S235/236), by ETP-47037 and ETP-47228 demonstrating efficient inhibition of the pathway by both ETP compounds underlying their specificity (**new Supplementary Fig. 1**). Moreover, the results obtained with the diverse set of PI3K inhibitors used in the current manuscript (see Figure 2), as well as the genetic validation for PI3K- α (see Fig. 3B,C), clearly demonstrate the link between PI3K/AKT inhibition and TRF1 modulation. Therefore, we consider that it is substantiated that TRF1 reduction is attributable to PI3K/AKT pathway inhibition.

[REVIEWER] The data generated with GSK3 inhibitors are impossible to interpret: GSK3i rescues the TRF1 signal reduction induced by AKTi, but not the reduction in protein levels. No mechanistic explanation is provided for this apparent paradox. Further adding to the confusion is the claim that AKT phosphorylates TRF1 and GSK3, but GSK3 does not phosphorylate TRF1. Yet, inhibiting GSK3 can reverse the effect of AKT on TRF1. How could these results be integrated into a coherent model for AKT-dependent regulation of telomeric function remains unclear. The role of GSK3 in WNT signalling is not mentioned in the manuscript.

[AUTHORS] We agree with reviewer that data on GSK3 doesn't add any clarifying mechanisms and we have removed these data in the revised manuscript as suggested by both reviewers.

Specific comments:

[REVIEWER] Fig. 1C: The data for ETP-47228 inhibition of Ser473 AKT phosphorylation (and the negative effect of ETP-50952 counterpart) are weak relative to ETP-47037. At 24h and 10 μ M ETP-47228 did not inhibit at all Ser473 AKT phosphorylation, which is also in contrast with the panel of commercial inhibitors shown in Fig. 2C. It is unlikely that the results obtained with the ETP-47228 can be attributed to PI3K/AKT inhibition. In order to convincingly show inhibition of PI3K/AKT pathway, phosphorylation of other downstream targets should be examined.

[AUTHORS] As it was mentioned before, ETP-47228 (10 μ M) is able to inhibit Ser473 AKT phosphorylation clearly at 4h similarly to ETP-47037. The reviewer is right if we consider this time point in the experiment. Nevertheless, it is clear the differential effect of ETP-47228 vs its negative control ETP-50952 at 1h/4h, pointing to this event as the responsible of the observed TRF1 modulation. As stated above, we have performed a more thorough time course inhibition profile including intermediate time points between

4h and 24h of AKT phosphorylation at Ser473 by ETP-47037, ETP-51259, ETP-47228 and ETP-50952 at 10 μ M in CHA-9.3 cell line. We include these new data in the revised manuscript (**new Fig. 1C; new Supplementary Fig. 1**). The results show that the inhibitory activity of ETP-47037 was more prolonged than that of ETP-47228. Thus, ETP-47037 mediated p-AKT S473 inhibition was maintained during the 24-hour treatment while the ETP-47228 mediated inhibition decreased 35% after 24 hours as compared to 1 hour treatment (**new Fig. 1C**). ETP-47228 (10 μ M) is able to inhibit Ser473 AKT phosphorylation clearly at 4h similarly to ETP-47037. It is reasonable that ETP-47228-mediated inhibition is sufficient to exert its effect on TRF1 modulation after 24h. As suggested by the reviewer, we have also performed a time course inhibition profile of two PI3K downstream effectors, PRAS40 (T246) and P6 (S235/236), by ETP-47037 and ETP-47228 demonstrating efficient inhibition of the pathway by both ETP compounds underlying their specificity (**new Supplementary Fig. 1**). Moreover, the results obtained with the diverse set of PI3K inhibitors used in the current manuscript (see Fig. 2), as well as the genetic validation for PI3K- α (see Fig. 3 B,C), clearly demonstrate the link between PI3K/AKT inhibition and TRF1 modulation. Therefore, we consider that it is substantiated that TRF1 reduction is attributable to PI3K/AKT pathway inhibition.

[REVIEWER] In Fig. 1D: Does 'PI3K activity' under the graph refer inhibition of PI3K activity or its inhibition? Same for the graph in Fig. 2D.

[AUTHORS] It refers to PI3K inhibition. This has been clarified in the revised manuscript.

[REVIEWER] Fig. 7A: MW markers should be included here because it is unlikely that the bands shown correspond to full length TRF1 (approx. 66 KDa) and GSK3 peptide 13 amino acid long (less than 2 KDa).

[AUTHORS] We have separated this figure into two different panels in **revised Fig. 5**. We also include the Molecular Weight (MW) markers in the revised figure.

[REVIEWER] Phosphorylation of the GSK3 peptide appears independent of AKT, because AKTi does not abrogate it. What is the authors explanation of this result? Whether GSK3 can phosphorylate TRF1 should also be addressed in this experiment, in order to demonstrate the specificity of the effect to AKT.

[AUTHORS] In the previous figure, we used 1.5 μ g of the GSK3 peptide used as a positive control in the reaction mix. We have repeated this experiment using a lower amount of the GSK3 peptide control (100 ng). In addition, we performed the reaction in the presence of 25 μ M and 2.5 μ M of MK-2206 (AKTi). The new data show that AKTi does also inhibit the phosphorylation of the GSK3 control peptide (see **new Fig. 5A**).

[REVIEWER] Fig. 7B,C: This figure is confusing; the graphs should be labelled with the conditions of which experiment is done. Figure legend is incorrect.

[AUTHORS] In the revised **Fig. 5B-E** (previous Fig. 7B-E), we have simplified the data shown by only including the LC/MS/MS corresponding to samples containing only TRF1 and TRF1&AKT.

[REVIEWER] Fig. 8: The authors are examining here a dominant negative phenotype, since endogenous TRF1 is expressed in the transduced cells (p53^{-/-} MEFs and CHA 9-3 mouse cancer cells). A meaningful experiment would be to transduce these

constructs into MEFs carrying a conditional *Trf1* deletion. This would eliminate any possible interference of endogenous TRF1.

[AUTHORS] We appreciate reviewer's suggestion. In the revised manuscript we include the suggested experiment in which we transduced MEFs TRF1 lox/lox with the different mutants and then deleted endogenous TRF1 by Cre expression in order to rule out possible interference of endogenous TRF1 (**new Fig. 6E,G**). The new results confirm that the TRF1 single mutants T330A and S344A, as well as the TRF1 double mutant (T330A/S344A) show a significant decrease in the intensity of TRF1 telomeric foci compared to MEFs expressing wild-type TRF1 (**new Fig. 6F**). In contrast, no significant differences were detected between TRF1-T248A and wild-type TRF1 (**new Fig. 6F**). In addition, in order to address the ability of these mutant variants to rescue the severe proliferative defects associated to TRF1 deficiency, we have analyzed growth rate in *Trf1*-deficient MEFs expressing eGFP-tagged *Trf1*^{wt}, *Trf1*^{T248A}, *Trf1*^{T330A} and *Trf1*^{S344A} alleles (**new Fig. 6G**). Importantly, we found that cells expressing TRF1-T330A and TRF1-S344A proliferated at similar rate than the *Trf1*-deleted ones, indicating that these mutations render non-functional TRF1 variants. In contrast, cells expressing TRF1-T248A grew at similar rate than cells expressing TRF1 wild-type, indicating that TRF1-T248A variant is able to complement TRF1 deficiency (**new Fig. 6G**). These new results clearly indicate the importance of AKT-dependent TRF1 phosphorylation for cell viability *in vivo*. These results have been included in the manuscript text and Discussion (**see page 2, Abstract; page 16, 2nd paragraph; page 18, 1st paragraph; page 23, 1st paragraph**).

[REVIEWER] Fig. 9: In this figure the authors use BYL-719, a commercial PI3Ka inhibitor and not one of the 'proprietor' compounds used in the other figures; no explanation is provided.

[AUTHORS] BYL-719 is used because it is a PI3K-alpha specific inhibitor (**new Fig.8**). The Pi3K-alpha isoform as demonstrated in the manuscript is the responsible for the regulation of TRF1. Since BYL-719 is currently in clinical trials and not the proprietary inhibitors, which are in preclinical level. Therefore, the implications of the results reported in this manuscript could be translated directly to the clinic using the advanced Novartis compound.

[REVIEWER] Fig. 9: It is not clear that the 53BP1 and gH2AX quantification refers to telomere damage. The compound used here could introduce non-specific DNA damage anywhere in the genome.

[AUTHORS] We fully agree with the reviewer that the data presented in **new Fig. 8G** (γ H2AX positive cells) indicate genome wide DNA damage. However, we also addressed specific telomeric damage by analyzing the so-called TIFs (Telomere induced DNA damage foci) (see **new Fig. 8H**) using colocalization of γ H2AX and RAP1 (a telomeric protein). Our data clearly shows that the BYL719 PI3K inhibitor induces genome wide DNA damage, and that a significant part of the damage is localized at telomeric DNA in those PDXs that responded effectively to the BYL719 treatment.

[REVIEWER] Fig. 9. The co-localisation of RAP1 and gH2AX signal in 9G is minimal.

[AUTHORS] As stated in the previous point, we observed a genome wide effect in DNA damage by BYL719 treatment. Part of this damage is found at telomeres since there is a moderate but significant increase in the number of TIFs (**new Fig. 8H,I**).

REVIEWERS' COMMENTS:

Reviewer #1 (Remarks to the Author):

The authors have addressed several important issues and achieved to significantly improve their manuscript. However, the questions concerning the clinical relevance of the presented data still remains as the revised manuscript includes only a very limited number of samples and in fact, there is no clear correlation between the phosphorylation status of AKT (AKT-Ser473) and telomere length. As the authors were able to correlate untreated PDXs with the lowest basal p-AKT levels with the lowest telomeric TRF1 levels, it should be possible to screen a significant number of samples (biopsy before neo-adjuvant chemotherapy) from an independent cohort of patients with breast cancer for pAKT, TRF1 levels and telomere length. Hence, my recommendation is major revision.

Reviewer #2 (Remarks to the Author):

I am satisfied with the manner in which the authors have addressed my comments and i recommend the paper for publication.